# Thermal modeling of three lakes within the continuous permafrost zone in Alaska using LAKE 2.0 model

Jason A. Clark[1], Elchin E. Jafarov[2,3], Ken D. Tape[1], Benjamin M. Jones[4], and Victor Stepanenko[5,6]

[1]Geophysical Institute, University of Alaska Fairbanks, AK, USA
[2]Earth and Environmental Sciences Division, Los Alamos National Laboratory, Los Alamos, NM, USA
[3]Woodwell Climate Research Center, Falmouth, MA, USA
[4]Institute of Northern Engineering, University of Alaska Fairbanks, AK, USA
[5]Lomonosov Moscow State University, Moscow, Russia
[6]Moscow Center for Fundamental and Applied Mathematics, Moscow, Russia

*Correspondence to*: Jason A. Clark (jaclark2@alaska.edu)

# Thermal modeling of three lakes within the continuous permafrost zone in Alaska using LAKE 2.0 model

**Abstract.** Lakes in the Arctic are important reservoirs of heat with much lower albedo in summer and
greater absorption of solar radiation than surrounding tundra vegetation. In the winter, lakes that do not freeze to their bed have a mean annual bed temperature > 0 °C in an otherwise frozen landscape. Under climate warming scenarios, we expect Arctic lakes to accelerate thawing of underlying permafrost due to warming water temperatures in the summer and winter. Previous studies of Arctic lakes have focused on ice cover and thickness, the ice decay process, catchment hydrology, lake water balance, and eddy
covariance measurements, but little work has been done in the Arctic to model lake heat balance. We applied the LAKE 2.0 model to simulate water temperatures in three Arctic lakes in Northern Alaska over several years and tested the sensitivity of the model to several perturbations of input meteorological variables (precipitation, shortwave radiation, and air temperature) and several model parameters (water vertical resolution, sediment vertical resolution, depth of soil column, and temporal resolution). The LAKE
2.0 model is a one-dimensional model that explicitly solves vertical profiles of water state variables on a grid. We used a combination of meteorological data from local and remote weather stations, as well as data derived from remote sensing, to drive the model. We validated modelled water temperatures with data of observed lake water temperatures at several depths over several years for each lake. Our validation of the LAKE 2.0 model is a necessary step toward modelling changes in Arctic lake ice regimes, lake heat
balance, and thermal interactions with permafrost. The sensitivity analysis shows us that lake water temperature is not highly sensitive to small changes in air temperature or precipitation, while changes in shortwave radiation and large changes in precipitation produced larger effects. Snow depth and lake ice strongly affect water temperatures during the frozen season which dominates the annual thermal regime of Arctic lakes. These findings suggest that reductions in lake ice thickness and duration could lead to more
heat storage by lakes and enhanced permafrost degradation.

## 1 Introduction

Approximately forty percent of Arctic lowlands are covered by lakes (Grosse et al., 2013). Lakes in the Arctic are important reservoirs of heat (Williamson et al., 2009) that affect permafrost thaw and carbon and methane emissions (Rowland et al., 2011; Abnizova et al., 2012). Lake water temperatures regulate heat
fluxes, biogeochemical activity, and are influenced by meteorological conditions and the surface radiative balance (Abnizova et al., 2012; Jeffries et al., 1999; Arp et al., 2011; Wik et al., 2016; Rouse et al., 1997). Increasing lake temperatures can thaw underlying permafrost, creating taliks and enhancing surface-groundwater interactions (Rowland et al., 2011; Jorgenson et al., 2006; Jorgenson and Shur, 2007). Understanding and modeling water temperatures in permafrost landscapes is critical to be able to predict
future talik development, permafrost thaw, and greenhouse gas releases (Grosse et al., 2013).
Multiple lake models attempt to capture temperature variability in Arctic lakes. Zero-dimensional models are used primarily in long timescale studies and have simplified numerical schemes allowing for computational efficiency (Mironov, 2008; Kirillin et al., 2011). The one-dimensional models have more sophisticated physical parametrization of the hydro- and thermo-dynamical processes and can effectively
simulate fine scale temporal (hourly, daily) thermal and biogeochemical processes. Recent two-dimensional models add advective heat transport through groundwater flow (Rowland et al., 2011; Grenier et al., 2018; Provost and Voss, 2019). Three-dimensional models couple three-dimensional surface flow to three-dimensional groundwater flow (Spanoudaki et al., 2009; Rueda and MacIntyre, 2010; Painter et al., 2016; Painter, 2011). Most high-fidelity physics models require an increase in computing time. Using high
temporal resolution meteorological data adds an additional burden to the compute time. Grant et al. (2021) stressed the important role of one-dimensional models serving as an optimal solution/tool when it comes to perform multiple runs for large numbers of lakes under different scenarios of climate change.

While much work has been done in the study and simulation of lakes in permafrost settings, few studies have addressed the sensitivity of arctic lakes to climate change. Langer et al. (2016) simulated two lakes in the Lena River Delta using a linked CryoGrid3 and FLake model using historic climate data and two climate projections (Representative Concentration Pathway 4.5 & 8.5) to assess the effects of shallow water bodies on permafrost degradation in land surface models. However, their analysis focuses on sediment temperatures, talik formation below the lakes, and formation of new waterbodies in thawing permafrost. Other studies have applied models to simulate hydrologic transport processes and pathways (Rueda and MacIntyre, 2010), carbon biogeochemistry and ebullition (Tan et al., 2015, 2017), lake thermal structure (Guo et al., 2021), and stratification and heat exchange (Boike et al., 2015) in Toolik Lake and other Arctic lakes. While several other models have been applied to simulate lake thermal dynamics in permafrost settings, they have focused on thermokarst shore expansion (Ling and Liao, 2016; Ling and Zhang, 2019), talik sediment temperatures, talik development, and refreezing (Ling and Zhang, 2003, 2004), lake ice phenology (Zhang and Jeffries, 2000; Morris et al., 2005), conductive heat flux through snow (Jeffries et al., 1999; Jeffries and Morris, 2006), or advective heat transport (Rowland et al., 2011) rather than lake water temperature, ice thickness and snow depth as in this study. Previous work has not addressed the sensitivity of lake water temperature, lake ice thickness or lake snow depth to changes in air temperature, precipitation, or shortwave radiation in Arctic Lakes.

We use the one-dimensional LAKE 2.0 model, which has been in active development since 2011, presenting a compromise between explicit resolution of key physical processes and computational efficiency (Stepanenko et al., 2011, 2016; Iakunin et al., 2020). This one-dimensional model allows high temporal resolution input data and computational efficiency combined with effective reproducibility of the thermodynamics of lakes. An advantage of the LAKE model in context of climate-lake-permafrost interaction studies, is that it explicitly simulates phase transitions in underlying ground at different depth zones of bottom sediments (Stepanenko et al., 2016).

The original version of the LAKE model was developed by Stepanenko and Lykossov, (2005) and Stepanenko et al., (2011) and then extensively validated against biogeochemical observations and similar types of models as a part of the Lake Model Intercomparison Project (Stepanenko et al., 2010, 2013, 2016). In recent studies, the main focus of the LAKE model developments was its biogeochemical module, which describes $O_2$, $CO_2$ and $CH_4$ gas exchange between the water column and sediments at different depths (Stepanenko et al., 2011; Iakunin et al., 2020). In spite of the considerable research testing the LAKE biogeochemical module, little has been done to validate water temperatures produced by the LAKE model for lakes in permafrost regions, rendering it difficult to predict the heat fluxes and impact to underlying frozen ground.

Here we validate the LAKE model thermodynamics using water temperature observations from three Arctic lakes in Alaska: Fox Den, Atqasuk, and Toolik. All three lakes are located within the continuous permafrost zone in the northern part of Alaska (Jorgenson et al., 2008; Obu et al., 2019). The lakes represent three different climate regimes spanning the continuous permafrost zone, ranging from -6 to -12 °C mean annual air temperatures. The morphometry of the lakes varies from small and shallow (Fox Den) to large and deep (Toolik). Validating lake water temperatures among diverse climatological conditions and lake morphometries in the Arctic allows us to quantify the robustness of the LAKE model. This is a crucial step toward applying the LAKE model to examine the impact of water bodies on permafrost.

## 2 Methods

### 2.1 Description of the model

The LAKE 2.0 is a 1-D model of thermodynamic, hydrodynamic and biogeochemical processes in the water basin and the lake bottom sediments (Stepanenko et al., 2016). The model uses the generic form of the Reynolds-averaged advection–diffusion equation applied to horizontal velocity components,

temperature, turbulent kinetic energy (TKE), TKE dissipation, and concentration of multiple
biogeochemical species such as gases $O_2$, $CO_2$ and $CH_4$ and organic carbon variables. The lake
thermodynamics includes a heat diffusion formulation where heat conductance is a sum of molecular and
turbulent coefficients computed involving the k-ε model (Stepanenko et al., 2016). The lake
hydrodynamics portion employs momentum equations and uses the Coriolis force for lakes with a
horizontal size that exceeds the internal Rossby deformation radius (Patterson et al., 1984). The model
takes into account heat and gas exchange of water column with a sloping bottom. The scheme for water
temperature and gas concentrations is coupled to sediment columns originating at the bottom at different
depths. The heat exchange in the sediment layer includes vertical transport and phase transition between
water and ice (Stepanenko et al., 2019). The vertical heat transport in sediments is described according to
Côté and Konrad (2005). Liquid water is transported via gravity and capillary-sorption forces (Stepanenko
and Lykossov, 2005). The snow and ice are represented with multilayer models, where heat conduction
with shortwave radiation absorption are taken into account (Stepanenko et al., 2019); in addition, in snow
module, the gravitational infiltration of water melted from the snow surface is simulated. The geothermal
heat flux at the lower boundary of the sediment layer is assumed to be zero. More detailed information
about mathematical formulations used in the model can be found in Stepanenko and Lykossov (2005),
Stepanenko et al. (2016) and Stepanenko et al. (2020).

**2.2 LAKE model setup**

The simulation conducted in the present study spans 2, 4, and 4 years at three study lake sites (Atqasuk,
Fox Den, and Toolik) respectively, with a 1 hour time step for the input and output data (see Appendix for
comparison of simulations using 1 hour input data to 1 day input data). In the set-up stage, specific features
of the study lakes were prescribed, namely the depth of the lake, area of lake surface, morphometry of the
lake bottom, the vertical water grid resolution, the vertical soil grid resolution and depth, and soil type
(Table 1). The LAKE model was initialized with water column temperature data measured at each site. For
each site, the model was spun up using 1 year of meteorological data repeated for 10 years (Table 1). The
resulting water and soil temperatures were used to initialize water and soil temperatures for the simulations
(Table 1). Atmospheric forcing input data were taken from local meteorological stations (Toolik, Atqasuk),
remote meteorological stations (Fox Den) and remotely sensed satellite data (Fox Den, Atqasuk).

**Table 1. Parameters of baseline lake simulations.**

| Lake name | Atqasuk | Fox Den | Toolik |
|---|---|---|---|
| Latitude | 70.452497 | 66.55877 | 68.631496 |
| Longitude | -156.951984 | -164.45670 | -149.607404 |
| Area, m2 | 2,732,050 | 17,861 | 1,492,898 |
| Time span of integration | 2013-08-12 to 2015-08-10 | 2009-06-10 to 2013-06-10 | 2013-05-17 to 2016-09-18 |
| Time step, seconds | 20 | 20 | 20 |
| Maximal lake depth, m | 2.54 | 1.5 | 26.0 |
| Vertical grid water column | 40 layers, refined near boundaries | 40 layers, refined near boundaries | 40 layers, refined near boundaries |
| Vertical resolution water column, m | 0.0635 | 0.0375 | 0.65 |
| Initial temperature at bottom of soil column, C | 5.0 | -4.0 | 4.0 |
| Calibrated temperature at bottom of soil column, C | 2.0 | -1.0 | 3.6 |
| Sediment (soil) type | Silt loam | Silt loam | Silt loam |

| | | | |
|---|---|---|---|
| Depth of soil column, m | 10.0 | 10.0 | 10.0 |
| Vertical resolution of soil column, m | 1.0 | 1.0 | 1.0 |
| Number of columns of sediments | 5 | 5 | 5 |
| Vertical grid in columns of sediments | 10 layers, exponentially compacting towards sediments top | 10 layers, exponentially compacting towards sediments top | 10 layers, exponentially compacting towards sediments top |
| Albedo for visible radiation | 0.06 | 0.06 | 0.06 |
| Fraction of near-infrared energy in shortwave flux | 0.35 | 0.35 | 0.35 |
| Water surface emissivity | 0.98 | 0.98 | 0.98 |
| Extinction coefficient for shortwave radiation, m-1 | 0.58 | 0.58 | 0.58 |
| Modal wind fetch, m | 1000 | 1000 | 1000 |

### 2.3 Input data

Since the LAKE model input files require a certain data format, we developed pre-processing scripts to streamline the LAKE model's input data. For Atqasuk, snow depth (m) was converted to precipitation (m s$^{-1}$, method described below). For Fox Den and Atqasuk incoming longwave and shortwave radiation were

140 taken from NASA CERES for the 1° x 1° grid cell containing the study lake (Wielicki et al., 1998; Rutan et al., 2015; Kato et al., 2018). Meteorological variables used as inputs included: wind speed and direction converted to 2-component wind vectors (m s$^{-1}$), air temperature converted to degrees Kelvin, atmospheric pressure converted to pascals, incoming longwave and shortwave radiation converted to W m$^{-2}$, humidity converted to kg kg$^{-1}$, and precipitation converted to m s$^{-1}$.

### 2.4 Data and script availability

Weather data and model infiles for simulations in this study have been archived and are publicly available (Jafarov et al., 2021). Observed water temperature data has been published and archived for Atqasuk (Hinkel et al., 2012), Fox Den (Jones et al., 2021), and Toolik Lake (MacIntyre and Cortes, 2017; Clark and Jafarov, 2021). Pre-processing scripts were developed to combine data sources, convert units, and

150 format meteorological data for input into the LAKE model. Post-processing scripts were developed to compare LAKE modeled water temperature to observed water temperature and to calculate model error. All processing scripts have been archived and are publicly available (Clark and Jafarov, 2021).

### 2.5 Study site: Atqasuk Lake

Atqasuk Lake 201 (70.452497, -156.951984) is located on the North Slope of Alaska, approximately 90 km

south of Utqiaġvik, AK. It is a large shallow lake (2,732,050 m$^2$) with a maximum depth of 2.54 m surrounded by sedge, moss, dwarf-shrub wetland tundra (Walker et al., 2005). The area is classified as continuous permafrost, but the presence and depth of permafrost under the lake is not confirmed (Jorgenson et al., 2008; Obu et al., 2019). Meteorological data was measured locally at the lake except for frozen season precipitation. Mean annual air temperature was -8.98 °C. Validation water temperatures were

measured hourly at 0.3 m and 2.5 m for 2013 to 2015 (Hinkel et al., 2012). The simulation period was 2013-08-12 to 2015-08-10 (Figure A1).
We pre-processed meteorological data for the Atqasuk lake from two sources: the South Meade USGS meteorological station and the Atqasuk lake meteorological station (Urban, 2017; Hinkel et al., 2012). The local meteorological data included: atmospheric pressure [mb], rainfall [mm], air temperature [°C], wind

speed [m s$^{-1}$], gust speed [m s$^{-1}$], wind direction [degree], and short and longwave radiation [W m$^{-2}$]. The

Atqasuk lake meteorological station did not have frozen season precipitation data. For frozen season precipitation we used hourly snow depth data from the South Meade USGS station (Urban, 2017). We extracted frozen precipitation from hourly snow depth by calculating the difference between the original and lagged snow depth time-series. Only positive differences were used for frozen precipitation. Due to sonic ranger instrument noise and wind-blown snow, the differenced time series contained high frequency small positive values that did not correspond to actual precipitation amounts. To filter this noise from the precipitation signal, we experimented with different minimum frozen precipitation threshold values and found that filtering precipitation values to remove values < 2.5 cm provides the best to match the observed lake water temperatures.

**2.6 Study site: Fox Den Lake**

Fox Den Lake (66.55877, -164.45670) is located on the Northwest portion of the Seward Peninsula in Western Alaska. It is a small lake (17,861 m$^2$) approximately 2 km from the Chukchi Sea coast, located in a drained lake basin with a maximum depth of 1.6 m, and surrounded by tussock sedge, dwarf-shrub, moss tundra (Walker et al., 2005). The area is classified as continuous permafrost, but the presence and depth of permafrost under the lake is not confirmed (Jorgenson et al., 2008; Obu et al., 2019). Meteorological data was not available locally at Fox Den. Instead, meteorological data from the National Weather Service station (Smith et al., 2011) at Kotzebue, AK (~90km ENE, station ID: 70133026616) was used and short and longwave radiation was obtained from NASA CERES (Wielicki et al., 1998; Rutan et al., 2015). Mean annual air temperature at Kotzebue was -5.37 °C for 2009 to 2013. Validation water temperatures were measured hourly at 1.5 m for 2009-2013 (Jones et al., 2021). The simulation period was 2009-06-10 to 2013-06-10 (Figure A2).

**2.7 Study site: Toolik Lake**

Toolik Lake (68.63150, -149.60740) is located on the North Slope of Alaska. It is a large and deep lake (1,492,898 m$^2$) with a maximum depth of 26.5 m surrounded by non-tussock sedge, dwarf-shrub, moss tundra and prostrate dwarf-shrub, herb tundra (Walker et al., 2005). The lake has a seasonal inlet and outlet. The area is classified as continuous permafrost, but the presence and depth of permafrost under the lake is not confirmed (Jorgenson et al., 2008; Obu et al., 2019). Meteorological data is collected at the nearby (~30 m W) Toolik Field Station and includes atmospheric pressure [mb], all season precipitation [mm], snow depth [m], air temperature [°C], wind speed [m s$^{-1}$], wind direction [degrees], and short and longwave radiation [W m$^{-2}$] (Edgar et al., 2018). Mean annual air temperature at Toolik was -7.36 °C. The simulation period was 2013-05-17 to 2016-09-18 (Figure A3).
Validation water temperatures were measured at 5 min intervals for 2013-2016 (MacIntyre and Cortes, 2017). The temperature sensors at Toolik lake were placed at 24 depths from 0 m to 20 m for the ice-free season (June - August) and at 19 depths from 3 m to 22 m for the ice season (September - May). For shallow depths (0 - 2 m), measured temperature was only available for the ice-free season, generally June through August. As water temperature measurement depths changed seasonally and annually, we interpolated the temperature data to hourly time intervals and 1 m depth intervals to compare with the model output data format. Inlet stream discharge and temperature were measured during the thaw season continuously using a pulse generator and stage-discharge relationships developed using periodic manual discharge measurements (Kling, 2019). Toolik inlet discharge data was formatted for the LAKE model but no processing or unit conversions were performed (Clark and Jafarov, 2021). Lake outlet discharge was assumed to be equal to inlet discharge for model input.
Toolik lake meteorological data was provided in a raw format with missing data and without error checking. For our purposes, Toolik lake meteorological data was gap-filled using a 7-day rolling average (Clark and Jafarov, 2021). Data was manually checked for errors and missing or erroneous values were replaced. Missing atmospheric pressure data for the winter of 2015/2016 was filled with data from winter 2014/2015. Missing precipitation values were filled with 0.

**2.8 Model Evaluation Metrics**

Lake simulations were compared to observations of lake water temperature using mean absolute error
(MAE), root mean squared error (RMSE), and bias calculated for each water depth available in the
observed data. Scenario data used in the model sensitivity analysis (section 2.12) was compared to baseline
simulations using the Z-score. Z-score is calculated as the mean of *Z's* for the scenario for a given model
variable (e.g. water temperature at 1 m depth), where each $Z$ is calculated as the difference of the model
variable, *x*, and the mean of baseline, μ, divided by the standard deviation of the baseline, σ, following Eq.
(1):

$$Z = \frac{x - \mu}{\sigma}. \tag{1}$$

**2.0 Model Water Resolution**

We investigated the effects of the vertical resolution of the water column on the accuracy of modeled water
temperatures using LAKE. We investigated both Atqasuk lake, a shallow lake (~2.5m) where the mixed
layer extends to the bottom of the lake, and Toolik lake, a deep lake (26m) where stratification can occur.
For Atqasuk we tested resolutions of 1.0m, 0.1m, 0.065m, and 0.025m. For Toolik we tested resolutions of
1.0m, 0.65m, 0.5m, and 0.25m.

**2.10 Model Soil Vertical Resolution**

We investigated the effects of the vertical resolution of the soil column on the accuracy of modeled water
temperatures using LAKE. We investigated Atqasuk lake using a 10m deep soil column and vertical
resolutions of 2.0m, 1.0m, 0.2m, and 0.1m.

**2.11 Model Temporal Resolution**

We investigated the effects of temporal resolution of the meteorological data on the accuracy of modeled
water temperatures using LAKE. We investigated Atqasuk lake and Fox Den lake using hourly and daily
data. For Atqasuk hourly data was averaged to daily data. For Fox Den an hourly and a daily dataset were
obtained from NOAA and NASA (see methods).

**2.12 Model Sensitivity Analysis**

To test the sensitivity of the model output to potential future weather conditions and to test the model's
sensitivity to inaccurate weather inputs, we altered the weather input for each lake using a common set of
240 scenarios. We choose the following scenarios: precipitation (P, -50%, -20%, -10%, +10%, +20%, +100%),
frozen precipitation (P_w, -20%, +20%), shortwave radiation (SW, -20%, -10%, +10%, +20%), and air
temperature (TA, -2 °C, -1 °C, +1 °C, +2 °C). The frozen season was defined as days with ice thickness >
0.0 m.

**3 Results**

**3.1 Model validation: Atqasuk Lake**

The Atqasuk modeled lake temperatures closely follow the observed temperatures at 0.3 m and 2.5 m over
the summer period (Figure 1). During the frozen season, the modeled temperatures underestimate cooling
in the lake (Table 2). There is mismatch towards the end of each frozen season which is likely explained by
ice rafting moving the temperature sensor into shallower water, which has been observed at many Arctic
lakes (Jones, personal communication). After ice-off and temperature sensor replacement, the model

captures summer temperatures well. A better match between modeled and observed temperature could be
achieved with more accurate snow accumulation data and a well anchored temperature sensor.

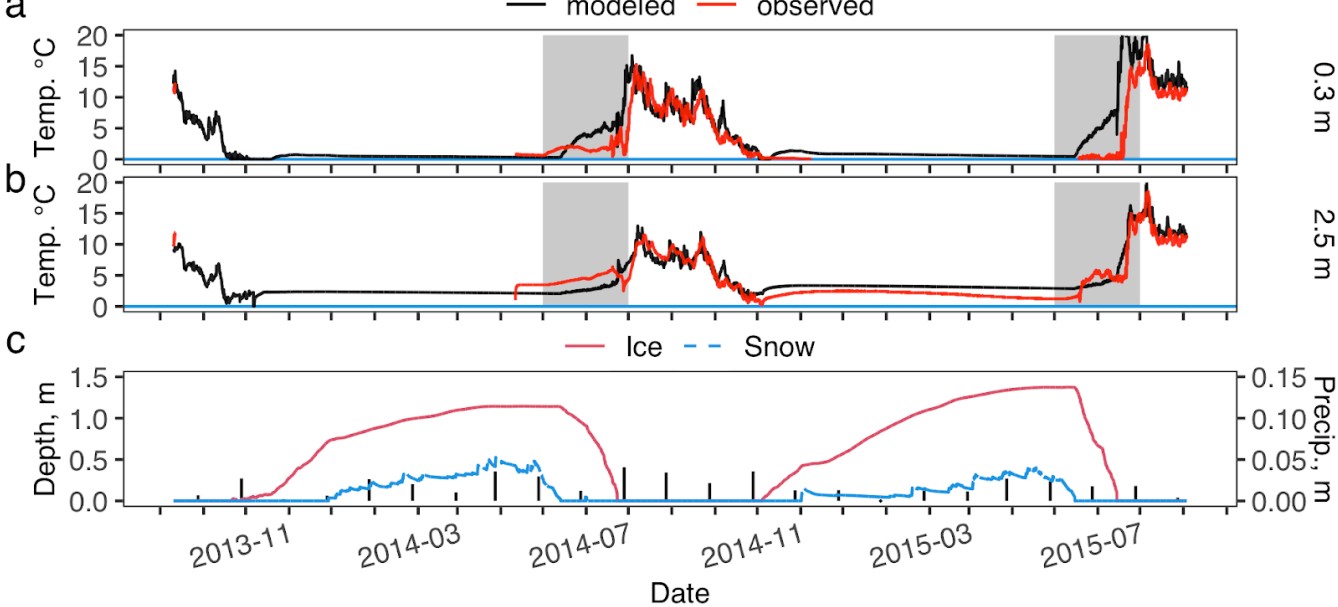

Figure 1. Atqasuk modeled and observed lake water temperature at 0.3 m (a) and 2.5 m (b) and modeled lake ice depth, lake snow depth
and measured monthly precipitation (vertical black bars) (c). Gray shading indicates periods of uncertainty in temperature sensor depth
when ice-rafting may have moved the sensor to shallower water. The y-axis of panel (a) is limited to water temperatures > 0 °C as the
LAKE model water temperature is limited to > 0 °C.

Table 2. LAKE model performance for Atqasuk, Fox Den, and Toolik lake water temperatures. Mean absolute error (MAE), root
mean squared error (RMSE) and bias (Bias) for total simulation period (annual), frozen, and thawed seasons. See methods for details.
The frozen season was defined as days with ice thickness > 0.0 m.

| | | | | | Season | | | | | |
|---|---|---|---|---|---|---|---|---|---|---|
| | | Annual | | | Frozen | | | Thawed | | |
| Lake | Depth | MAE | RMSE | Bias | MAE | RMSE | Bias | MAE | RMSE | Bias |
| Atqasuk | 0.3m | 5.07 | 7.15 | -4.74 | 6.16 | 8.09 | -6.01 | 2.74 | 4.54 | -2.02 |
| | 2.5m | 1.3 | 1.44 | 0.448 | 1.32 | 1.38 | -0.399 | 1.23 | 1.55 | -0.573 |
| | **All** | **3.18** | **5.16** | **-2.59** | **3.74** | **5.8** | **-3.2** | **1.99** | **3.39** | **-1.3** |
| Fox Den | 1.5m | 1.51 | 2.43 | -1.11 | 1.04 | 1.41 | -0.713 | 2.4 | 3.66 | -1.87 |
| | **All** | **1.51** | **2.43** | **-1.11** | **1.04** | **1.41** | **-0.713** | **2.4** | **3.66** | **-1.87** |
| Toolik | 1m | 1.12 | 1.29 | -1.01 | - | - | - | 1.12 | 1.29 | -1.01 |
| | 3m | 1.59 | 2.02 | 0.464 | 1.46 | 1.64 | 1.26 | 1.89 | 2.69 | -1.39 |
| | 5m | 1.78 | 2.24 | 0.964 | 1.73 | 1.98 | 1.35 | 1.87 | 2.7 | 0.15 |
| | 10m | 1.9 | 2.35 | 1 | 1.65 | 1.99 | 1.13 | 2.42 | 2.96 | 0.742 |
| | 19m | 1.77 | 2.19 | 1.07 | 1.65 | 1.99 | 1.32 | 2.01 | 2.56 | 0.527 |
| | **All** | **1.63** | **2.08** | **0.666** | **1.55** | **1.85** | **1.21** | **1.78** | **2.41** | **-0.235** |

**3.2 Model validation: Fox Den**

The modeled temperatures at Fox Den follow the observed temperatures (one depth: 1.5 m) over the
simulation period (Figure 2). There is some temperature mismatch during ice-off and during the thaw
season. As with Atqasuk, the temperature sensor at Fox Den was subject to ice raft induced movement

during ice-off. Additionally, more accurate modeled temperatures may have been achieved with local meteorological data and a well anchored temperature sensor.

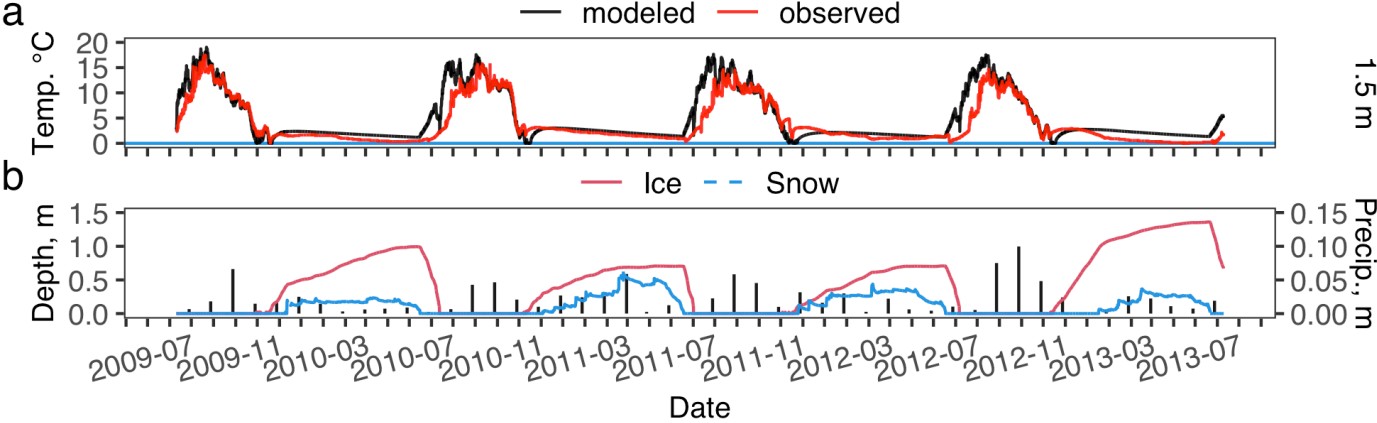

Figure 2. Fox Den Lake modeled and observed water temperature at 1.5 m (a) and modeled lake ice depth, modeled lake snow depth and measured monthly precipitation (vertical black bars) (b).

### 3.3 Model validation: Toolik Lake

Modeled water temperature was very similar to observed water temperature for Toolik Lake for all years (Figure 3 & 4). For the thaw seasons the modeled shallow (1, 3 m) water temperature was underestimated while deeper water temperature was overestimated (Table 2). For the frozen seasons the modeled water temperatures underestimated at all depths (Table 2). From 2015-08 through 2015-09 at 3m depth, the observed water temperature appears to be in error, as the 3m temperature departure is not seen at other depths. It was apparently corrected in mid- 2015-09 with the placement of the winter sensor array.

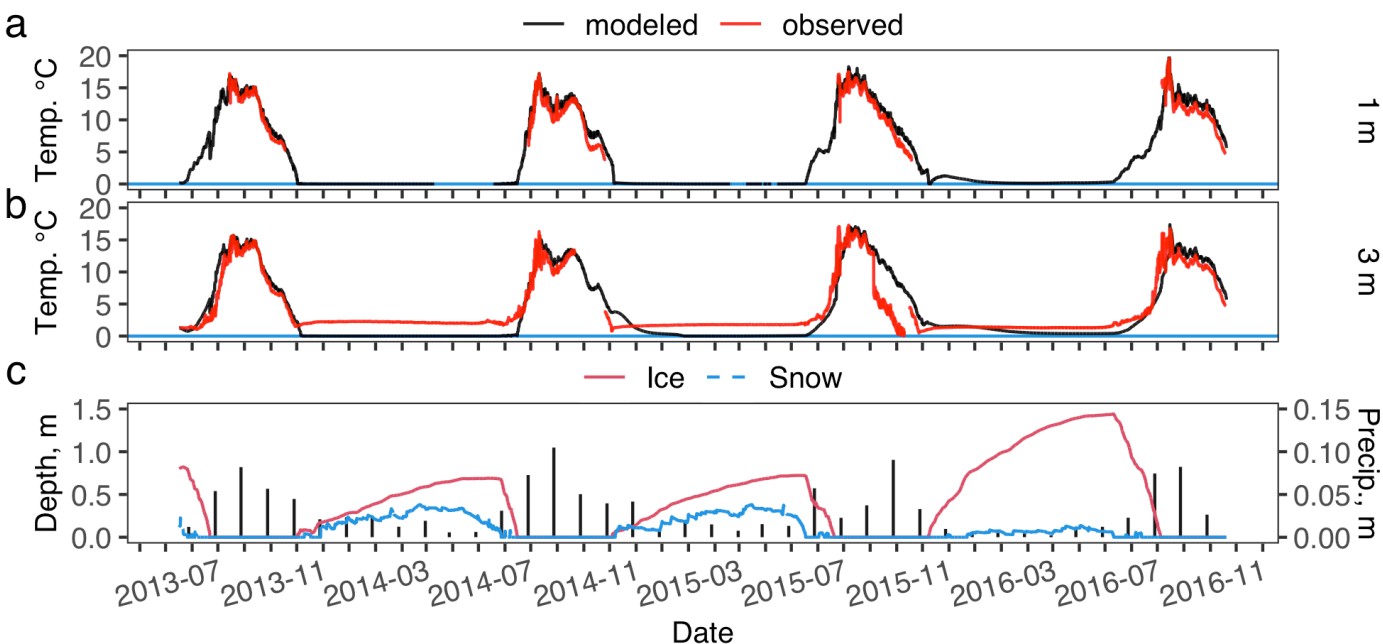

Figure 3. Toolik Lake modeled and observed lake water temperature at 1 m (a) and 3 m (b) and modeled lake ice depth, lake snow depth and measured monthly precipitation (vertical black bars) (c). Missing observed data in (a) and (b) is due to seasonal placement and removal of the temperature sensors, see Methods for details. The y-axis of panel (a) is limited to water temperatures > 0 °C as the LAKE model water temperature is limited to > 0 °C.

For 10 m and 19 m depths, modeled water temperature at Toolik tracked the general patterns of observed temperatures with some departures (Figure 4). For 10 m and 19 m summertime water temperature, the model only partially captures the observed temperature patterns while underestimating frozen season temperature in most years (Table 2).

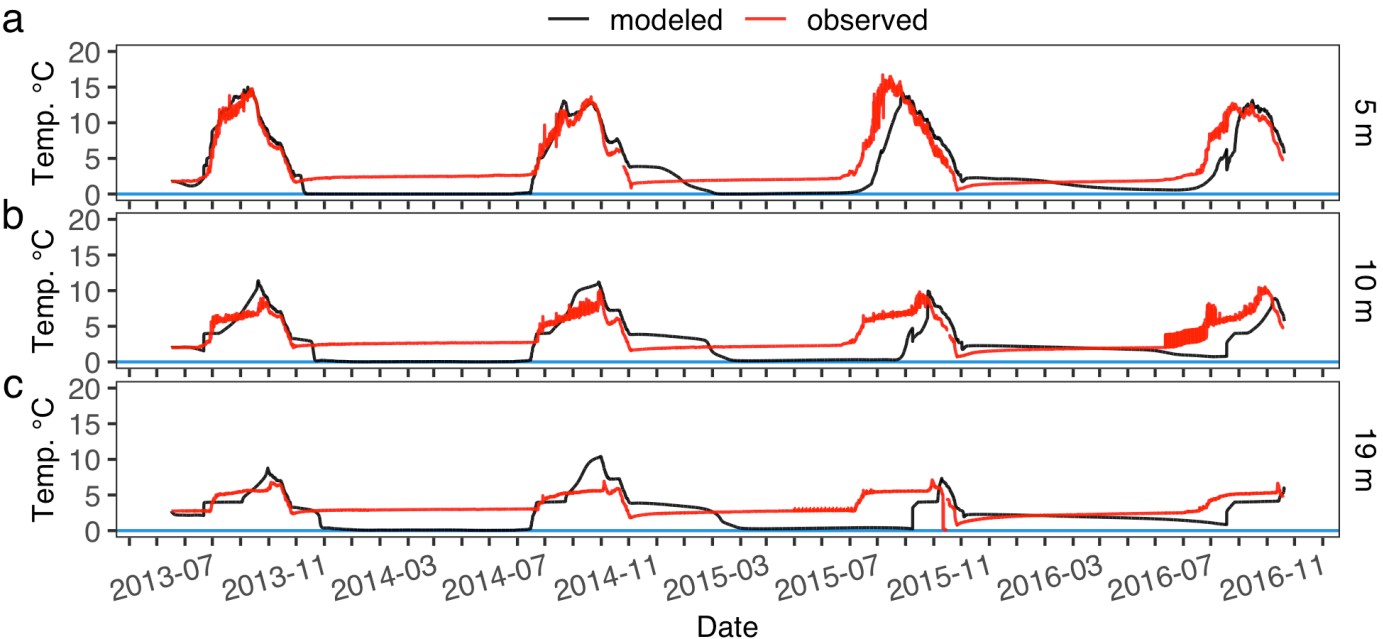

Figure 4. Toolik Lake modeled and observed lake water temperature at 5 m (a), 10 m (b) and 19 m (c).

### 3.4 Sediment temperature and heat flux

Lake sediment temperatures for all three lakes were modeled to a depth of 10 m (Figure C2). Observed sediment temperatures were not available for model validation. Shallow sediments (< 3 m) experience more warming in the shallow lakes (Atqasuk, Fox Den) as compared to the deeper lake (Toolik). All three lakes had near constant deeper (> 5 m) sediment temperatures over the several years of simulation. Downward heat flux at the bottom of the lake showed strong seasonal patterns of stronger flux during the thaw period and low fluxes during the frozen period (Figure C3). The shallower lakes (Atqasuk, Fox Den) had larger magnitude fluxes (compared to the deeper lake, Toolik) due to turbulent mixing in the thaw season.

### 3.5 Water Column Vertical Resolution

For Atqasuk 0.3m water depth, 1m water resolution had the lowest modeled water temperature error (RMSE=6.87) although higher resolutions were very similar (Figure D1 & Table D1). For Atqasuk 2.5m water depth, 0.1m, 0.065m, and 0.025m had the lowest modeled water temperature error (RMSE=1.44) although 1m resolution was slightly greater (RMSE=1.64, Figure D2 & Table D1). Vertical resolution of the water column had little effect on Atqasuk snow and ice layer thickness (Figure D3). For Toolik 1m and 3m water depths, 0.65m water resolution had the lowest modeled water temperature error (RMSE=1.29, RMSE=2.02) although higher resolutions were very similar for Toolik (Figure D4, Figure D5, & Table D2). For Toolik 5m, 10m, and 19m water depths, 1m water resolution had the lowest modeled water temperature error (RMSE=2.01, RMSE=1.84, RMSE=1.78), while 0.65m, 0.5m, and 0.25m resolutions were similar (Figure D6, Figure D7, Figure D8, & Table D2). Vertical resolution had little effect on Toolik snow and ice layer thickness (Figure D9).

### 3.6 Soil Column Vertical Resolution

For Atqasuk 0.3m water depth, 1m soil resolution had the lowest modeled water temperature error (RMSE=7.15) although higher resolutions were very similar (Figure E1 & Table E1). For Atqasuk 2.5m water depth, all soil resolutions had the same modeled water temperature error (RMSE=1.44). Vertical resolution of the soil column had little effect on snow and ice layer thickness (Figure E3).

**3.7 Temporal Resolution**

For both Atqasuk and Fox Den, hourly and daily temporal meteorological data resolutions had similar
water temperature errors, but daily resolution had a lower error value (Table F1). Temporal resolution had
little effect on lake temperatures (Figures F1, F2 & F3; Table F1). Daily temporal resolution appears to
reduce ice layer thickness (Figures F4 & F5). However, for Fox Den there is an apparent disagreement
between the daily and hourly precipitation data. Observed data are unavailable to compare to modeled
snow depth or ice thickness.

**3.8 Sensitivity analysis**

The sensitivity scenarios (TA, P, P_w, and SW) had minimal effect on water temperature (2.5 m depth) for
Atqasuk (-0.13 to 0.12 Z-score, Figure 5). For Fox Den, the effect on water temperature (1.5 m depth) was
minimal and similar to Atqasuk (-0.12 to 0.1 Z-score). For Toolik, there were slightly larger positive Z-
scores for shallow depth water temperatures (1, 3, and 5 m depth, -0.17 to 0.23) and also for deeper depths
(10 and 19 m depth, -0.03 to 0.33) compared to the other two lakes. For all three lakes, shortwave radiation
had the greatest effect on water temperature followed by air temperature and precipitation with a few
exceptions (Figure 5), most notably P_50% and TA-2C.
Ice thickness showed moderate effects from the sensitivity scenarios for Fox Den (-0.53 to 0.6), but lesser
effects for Atqasuk (-0.41 to 0.49) and Toolik (-0.12 to 0.3, Figure 5). The P_50% scenario had the greatest
effect on increasing ice thickness for all three lakes while the P_200% scenario had the greatest effect for
decreasing ice thickness but only for Atqasuk and Fox Den. Ice thickness is strongly influenced by snow
depth, particularly in the extreme precipitation scenarios.
Snow thickness was most strongly affected by the P scenarios for all lakes (Figure 5). As expected, the
extreme scenarios (P_50% and P_200%) had the greatest effect on snow depth. Increasing P, decreasing
SW, and decreasing TA all increased snow depth; decreasing P, increasing SW, and increasing TA all
decreased snow depth. Compared to year-round precipitation (P) scenarios, frozen season precipitation
(P_w) scenarios were similar for water temperature, ice thickness, and snow depth (Figure 5).

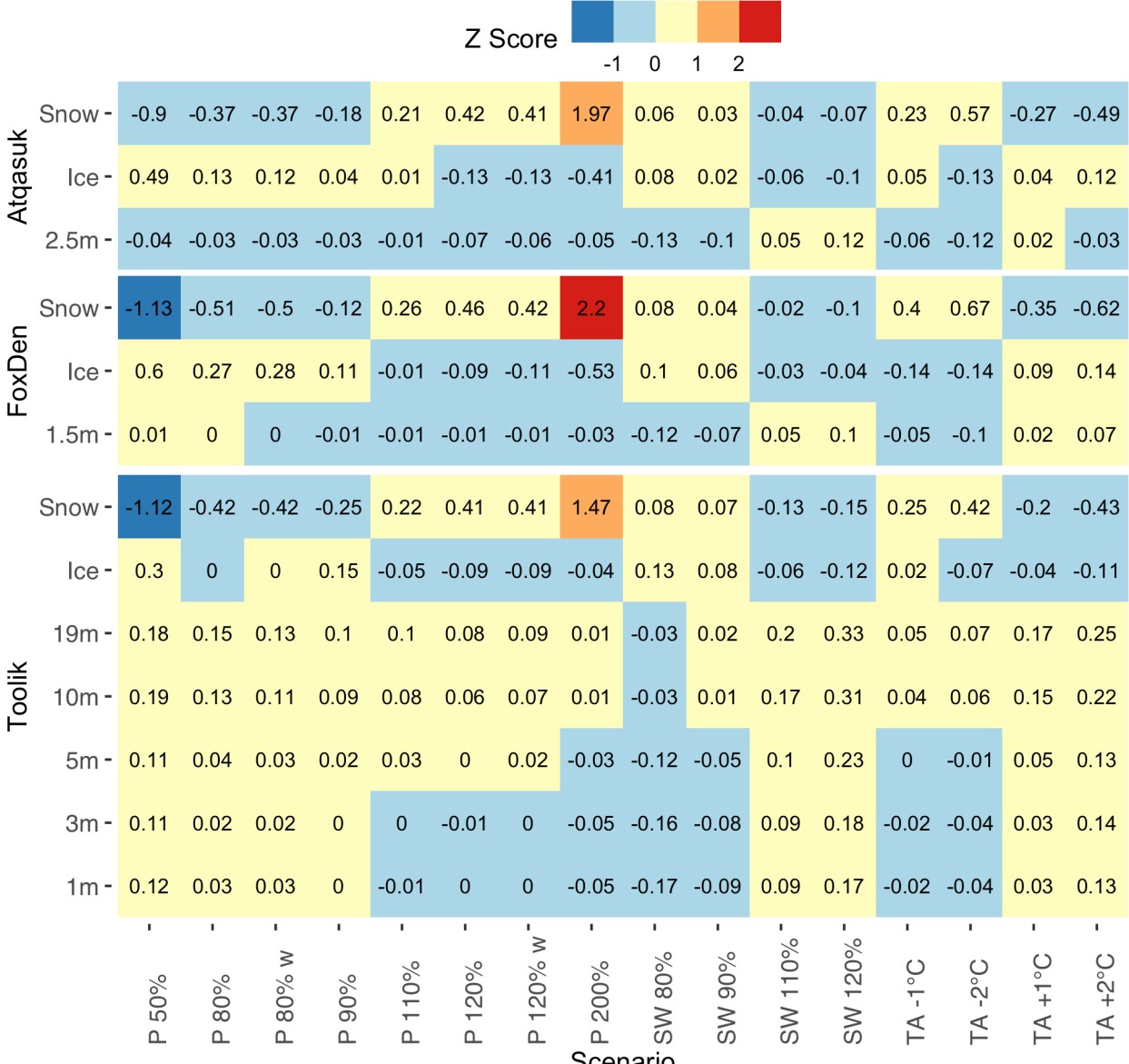

**Figure 5. Model sensitivity matrix showing mean Z-score for each scenario compared to the baseline model scenario for each response variable (ice thickness (m), snow depth (m), and water temperature at various depths). The sensitivity scenarios are: precipitation (P, -50%, -20%, -10%, +10%, +20%, +100%), frozen precipitation (P_w, -20%, +20%), shortwave radiation (SW, -20%, -10%, +10%, +20%), and air temperature (TA, -2 °C, -1 °C, +1 °C, +2 °C). Warm colors represent positive Z-scores, while cool colors represent negative Z-scores. Lighter colors indicate smaller Z-scores while darker colors indicate greater Z-scores.**

Focusing on the sensitivity of water temperatures during the frozen and thawed season, we find that most of the scenarios had a greater effect in the frozen season for the shallow lakes (Atqasuk, Fox Den), however for Toolik Lake there was a greater effect in the thawed season (Figure 6). Increased SW scenarios had positive Z-scores for all three lakes. In general, Z-scores for Atqasuk were negative for most scenarios (except SW increases), Z-scores were negative for the P scenarios and decreased SW and TA scenarios for Fox Den, while for Toolik Z-scores were negative for most scenarios (except decreased SW and P_200%). The larger effect of the P scenarios in the frozen season for Atqasuk may be attributed to snow cover thickness playing a large role in controlling frozen season water temperature of a shallow lake.

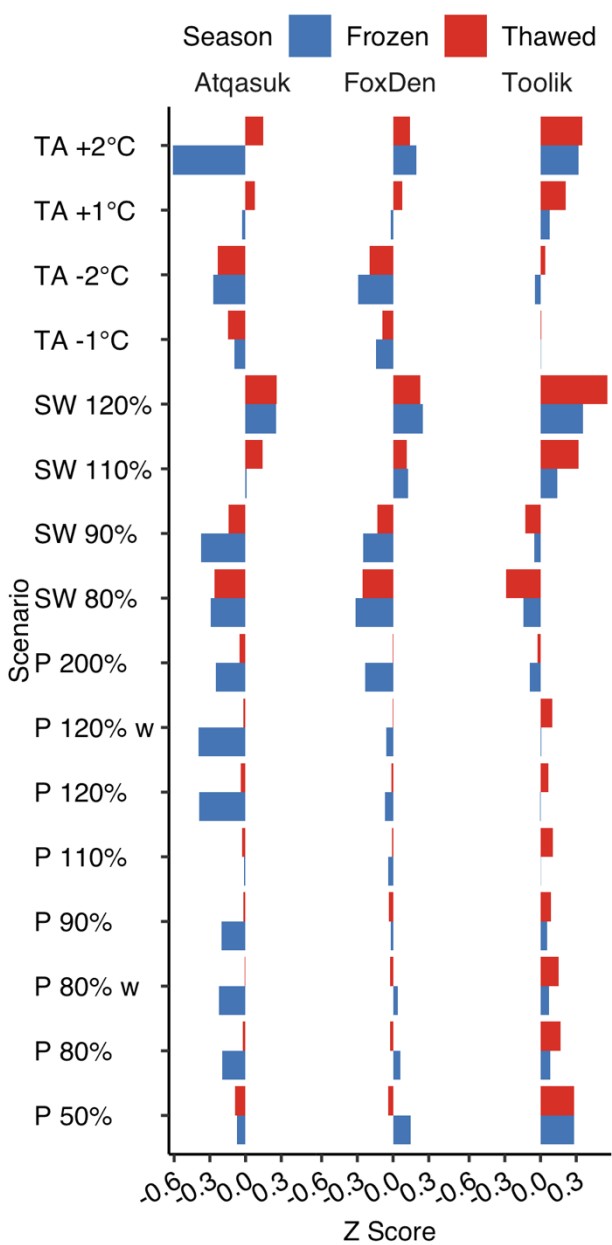

**Figure 6. Model water temperature response (mean z-score, all water depths) across the three lakes, 16 scenarios, and two seasons (frozen and thawed, based on ice presence) as compared to the baseline scenario. Blue bars are the frozen season, red bars are the thawed season. See Methods or Figure 5 for description of the scenarios.**

## 4 Discussion

### 4.1 Differences between study lakes

Despite large differences in lake size, shape, depth, and meteorology, the LAKE model performed well in reproducing lake water temperatures in all three study lakes. Toolik Lake is a large and deep lake compared to the shallow small Fox Den and the shallow but large Atqasuk lake. Toolik lake also has a large inlet and outlet stream which contributes substantial water influx and efflux during the thaw season. We modeled Toolik Lake without including inflow and outflow data but we were unable to achieve good matches with measured lake temperatures (Figs. B1 & B2). Most notably, water temperature mixing in the profile was delayed and weaker during the period of ice-off and throughout the summer. Modeled shallow water (1 m) temperature exceeded the observed temperatures and modeled deep water temperatures remained warmer than observed water temperatures during the frozen season and colder during the thawed

season. Incorporating inflow and outflow into the Toolik simulations improved the model fit in the thawed season, but not annually (Table B1) or in the frozen season (Table B2). Inflows after ice-off helped to promote vertical mixing of the temperature profile, while frozen season modeled temperatures went from too warm to too cold (Table B2).

The discharge data used for Toolik lake was only available for a limited period of time during the thawed season. Discharge measurements did not start until personnel arrived with discharge equipment (mid-May to early-June). Discharge measurements from the early thaw season (likely beginning in early-May) late thaw season and frozen season (September to May) are missing. This type of partial dataset highlights the need to collect data on ecological timeframes (i.e. the full thaw and full discharge season).

**4.2 Differences between terrestrial and lake based meteorological stations**

There was minimal mismatch between modeled lake water temperatures and measured lake water temperatures for all three lakes. The errors in modeled lake temperatures can be partially attributed to the quality of the meteorological data used to drive the LAKE model, particularly for frozen season precipitation. Furthermore, all meteorological data used in this study was collected from terrestrial stations (at various distances), not from the lake surfaces, further adding to potential modeled lake temperature error.

It is well known that meteorological conditions on lakes differ from that of the nearby terrestrial surfaces due to differences in surface albedo, heat exchange, surface roughness, and fetch. Furthermore, it is common for Arctic lake snow cover to differ from terrestrial snow cover (Sturm and Liston, 2003) as high winds can remove the complete snowpack from frozen lakes many times during the frozen season. In addition, much of the snowpack may be converted to snow ice, effectively reducing the snow thickness (Leppäranta, 2014). Snowpack plays an important role in the surface energy balance of lakes (Jeffries and Morris, 2006; Jeffries et al., 1999). Arctic lakes without snowpack would have substantially greater ice thickness and lower water temperatures throughout the frozen season (Alexeev et al., 2016). We do not have validation data for lake snowpack for our three study lakes. In future studies, it would be worthwhile to test the ability of the model to produce this phenomenon and ideally recreate it for a known lake with a validation dataset of water temperature, lake ice thickness, and lake snow depth.

**4.3 Observed LAKE model dynamics**

The Z-scores in water temperature were limited in the model by negative feedbacks between surface temperature and surface heat budget components. The rise of surface temperature by increasing air temperature (TA scenarios) or shortwave radiation (SW scenarios) leads to enhancing upward longwave radiation and evaporative heat losses, hindering further temperature rise. A similar effect of suppressing the surface temperature departure from baseline simulation takes place under the opposite air temperature and shortwave radiation deviations. The applicability of this modeling result to real-world processes is limited by not taking into account the lake effects on atmospheric state, and in a fully coupled lake-atmosphere system, similar sensitivity experiments should provide different estimates.

**5 Conclusion**

**5.1 Sensitivity analysis summary**

The sensitivity analysis shows us that the LAKE model is robust and not highly sensitive to the weather data perturbations used in this study. Even the relatively extreme scenarios (P_50%, P_200%, SW_120%, TA_+2, and TA_-2) produced only moderate water temperature Z-scores < 2.0. More moderate scenarios produced minimal Z-scores, < 1.0 in a majority of scenarios.

### 5.2 Local vs remote met data for LAKE model

Local meteorological data for the arctic is often lacking, of poor quality, or often contains large spans of missing data due to equipment failure and harsh weather conditions. Meteorological data for lakes is especially rare as the vast majority of meteorological stations are terrestrially located. We were encouraged by our Fox Den results that we were able to reasonably accurately model lake water temperatures with daily remote meteorological data and remotely sensed radiative fluxes. Future applications of the LAKE model will benefit from the abundance of daily meteorological data, remote meteorological data, and remotely sensed radiative fluxes.

### 5.3 Application of RS for LAKE model

Remote sensing has the potential to provide much needed local data for lake surface conditions including snow depth, ice thickness, and surface radiation fluxes. While we did not include remotely sensed snow depth or ice thickness in this study, such data would serve to better estimate and validate lake modeling efforts over large areas without local meteorological stations. However, care should be taken in applying remote sensing data without local validation datasets. The LAKE model could be used for the development of a reduced complexity model that could be applied in combination with the remotely sensed data to evaluate the thermal impact of widespread Arctic lakes on thawing permafrost.

### 5.4 Application of LAKE 2.0 in permafrost environments

LAKE 2.0 is a robust model that is appropriate for modeling the thermal interaction of surface waters and permafrost that is widespread in Arctic landscapes. To the best of our knowledge, the sensitivity of lake thermal and ice regimes to perturbations of atmospheric forcing in the continuous permafrost zone has not been addressed in the literature. Sub-lake permafrost and talik development below shallow lakes are topics of interest as the Arctic continues to experience unprecedented warming, shifting lake thermal dynamics and thawing permafrost (e.g. Peng et al., 2021; Langer et al., 2016; Boike et al., 2015; Alexeev et al., 2016; Creighton et al., 2018; Parsekian et al., 2019; Arp et al., 2016; Lin et al., 2010). Empirical studies have already shown that shallow lakes have warmed substantially over the last 30 years and may have already begun talik development (Arp et al., 2016). LAKE 2.0 can be a valuable tool to explore the thermal effects of new and developing arctic lakes on underlying permafrost. However, paired observational data of lake water and lake sediment temperatures (at several depths) are needed to validate simulations of lake sediment temperatures.

## 6 Appendices

**Appendix A: Meteorological data used in simulations**

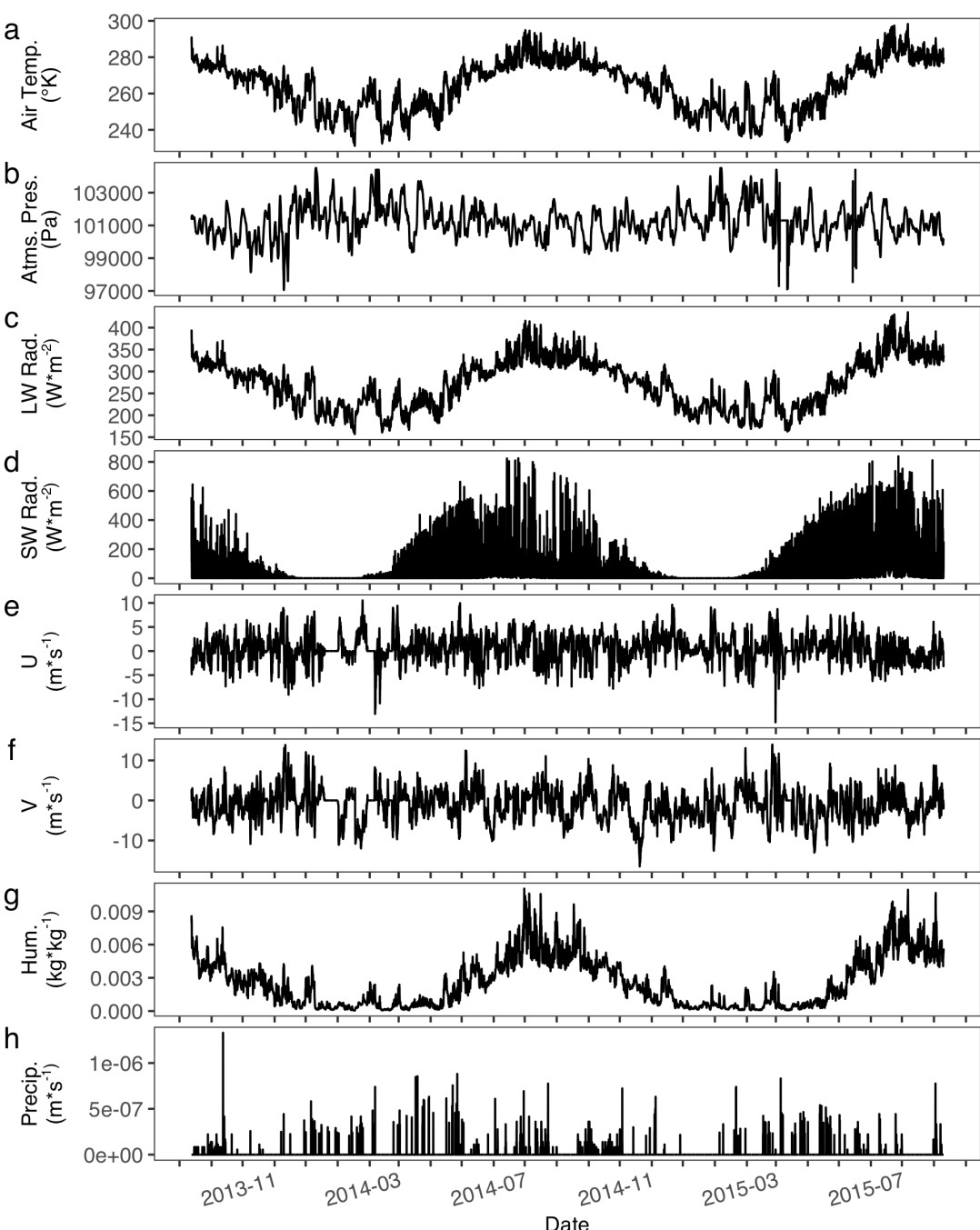

**Figure A1. Atqasuk hourly meteorological data used in simulations. Air temperature °K (a), atmospheric pressure Pa (b), longwave radiation W m² (c), shortwave radiation W m² (d), 2 component wind speed m s⁻¹ (e & f), humidity kg kg⁻¹ (g), and precipitation m s⁻¹ (h).**


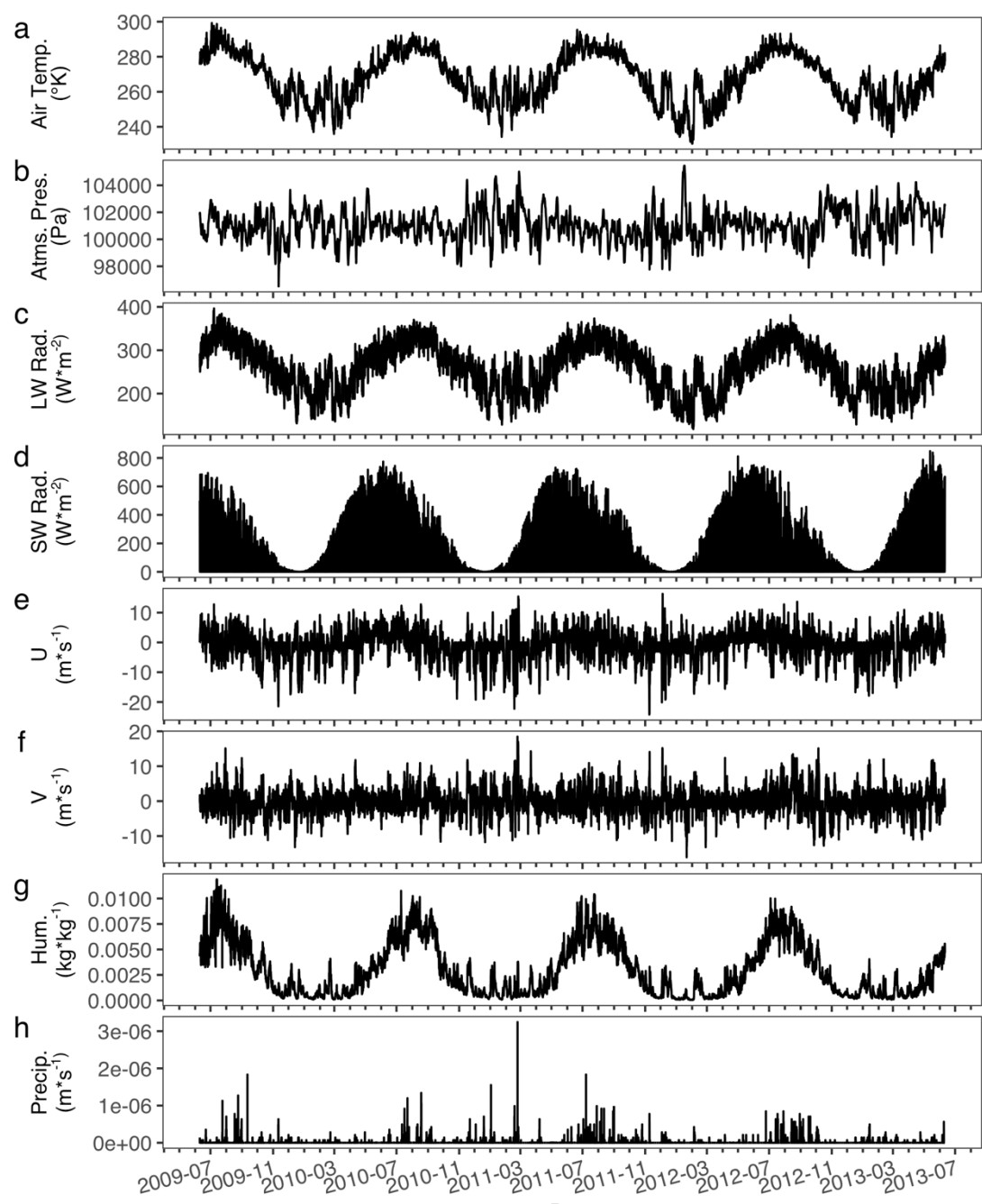

**Figure A2. Fox Den hourly meteorological data used in simulations. Air temperature °K (a), atmospheric pressure Pa (b), longwave radiation W m² (c), shortwave radiation W m² (d), 2 component wind speed m s⁻¹ (e & f), humidity kg kg⁻¹ (g), and precipitation m s⁻¹ (h).**


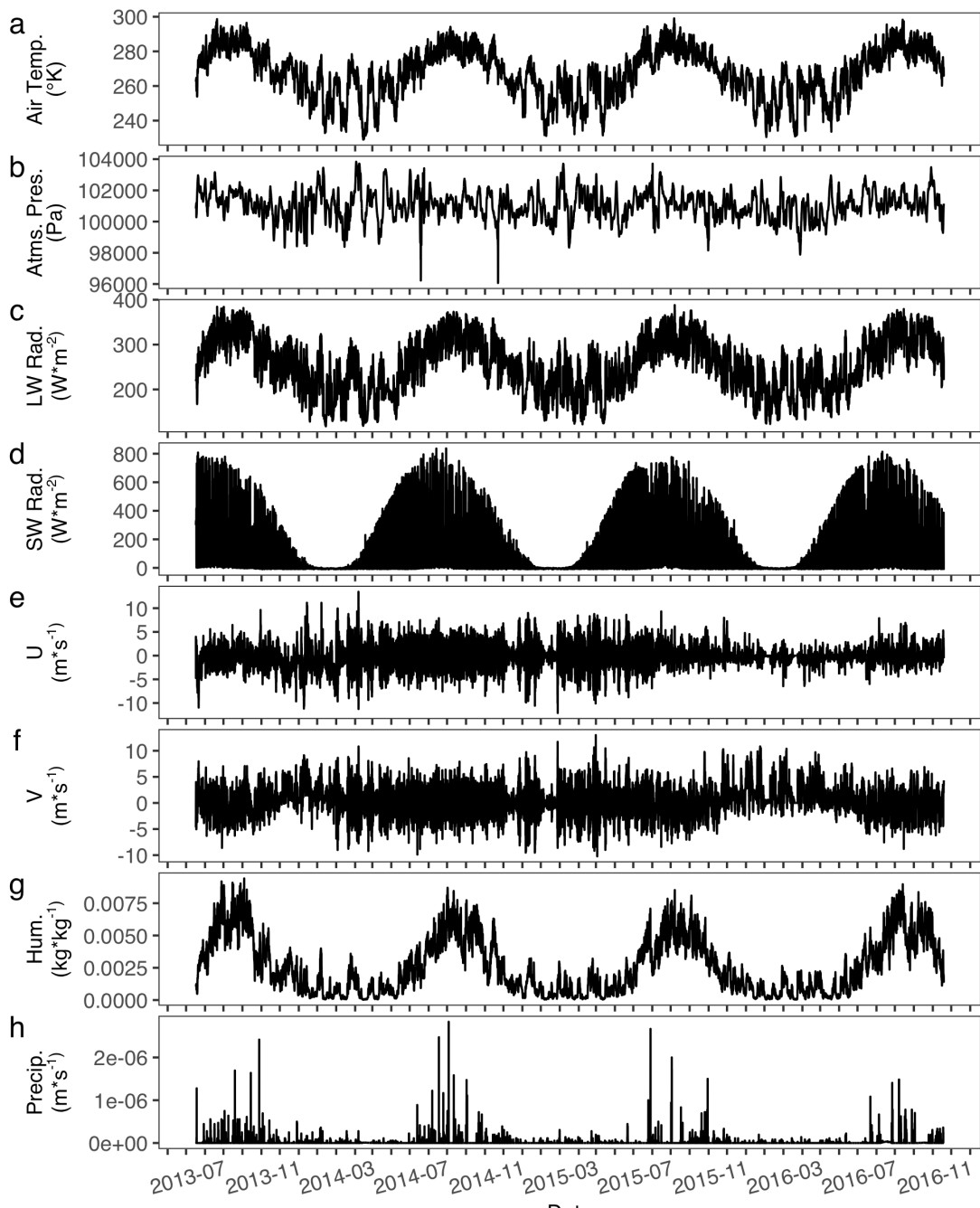

**Figure A3. Toolik hourly meteorological data used in simulations.** Air temperature °K (a), atmospheric pressure Pa (b), longwave radiation W m² (c), shortwave radiation W m² (d), 2 component wind speed m s⁻¹ (e & f), humidity kg kg⁻¹ (g), and precipitation m s⁻¹ (h).

**Appendix B: Toolik Lake simulations comparing simulations with and without inflow and outflow**

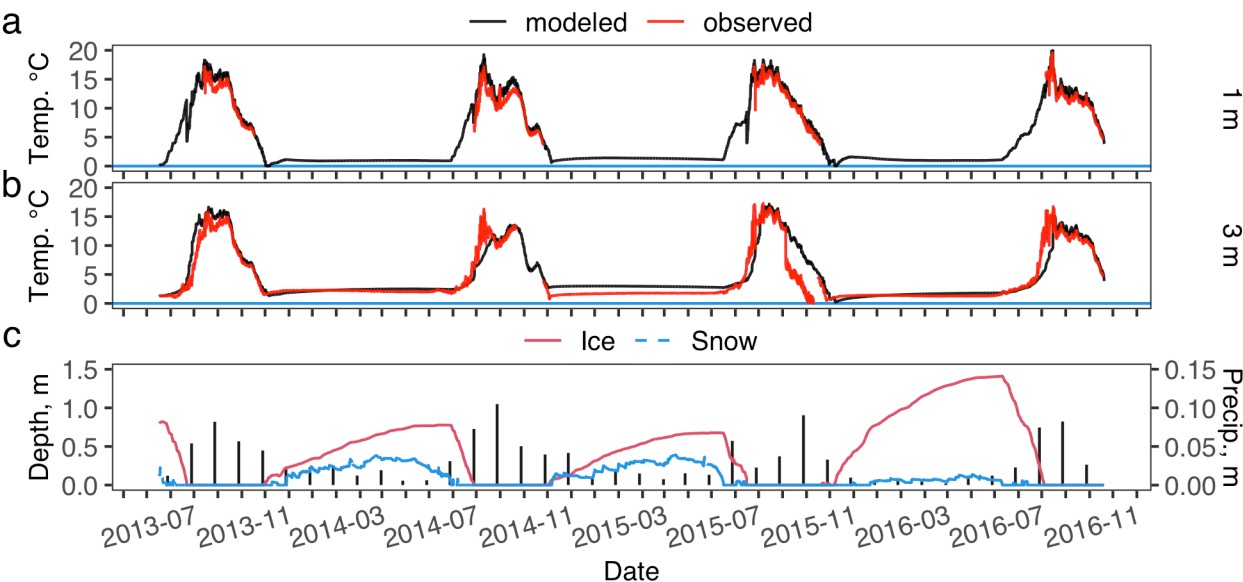

**Figure B1. Modeled and observed lake water temperature at 1 m (a) and 3 m (b) and modeled lake ice depth, lake snow depth and measured monthly precipitation (vertical black bars) (c) for Toolik lake with no inflow. Missing observed data in (a) and (b) is due to seasonal placement and removal of the temperature sensors, see Methods for details.**


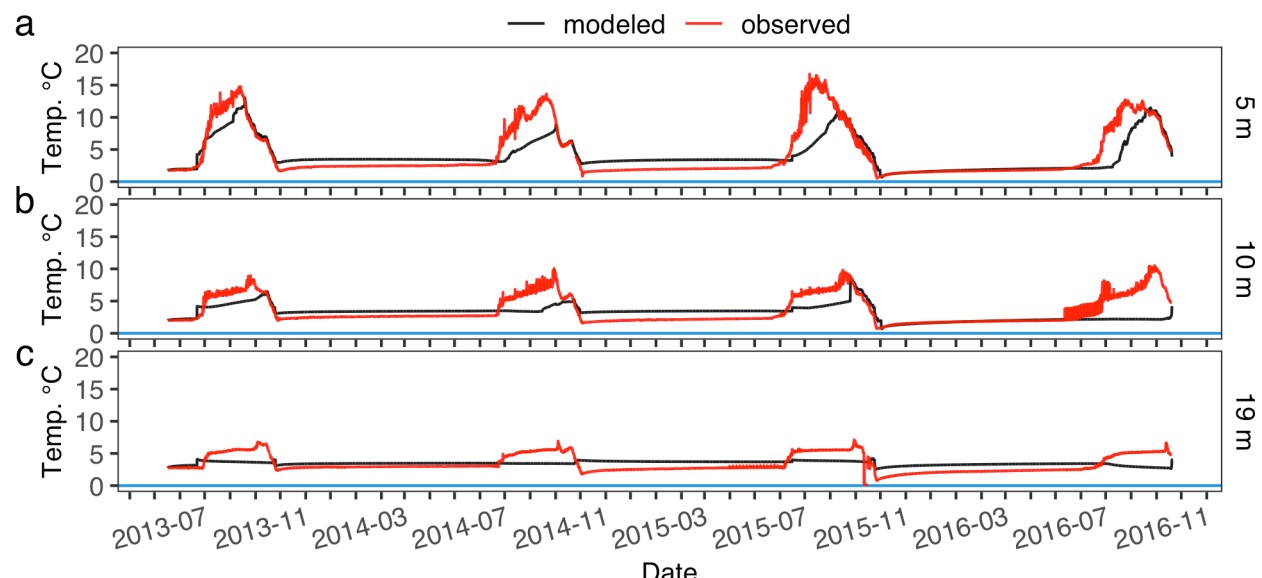

**Figure B2. Modeled and observed lake water temperature at 5 m (a), 10 m (b) and 19 m (c) for Toolik Lake with no inflow.**

**Table B1. Toolik model lake temperature error (RMSE) for inflow and no-inflow simulations.**

| Depth, m | Toolik inflow | Toolik no inflow |
|---|---|---|
| 1 | 1.29 | 1.22 |
| 2 | 1.13 | 1 |
| 3 | 2.02 | 1.71 |
| 4 | 1.91 | 1.81 |
| 5 | 2.24 | 2.29 |
| 6 | 2.44 | 2.61 |

| | | |
|---|---|---|
| 7 | 2.52 | 2.52 |
| 8 | 2.4 | 2.34 |
| 9 | 2.36 | 2.13 |
| 10 | 2.35 | 1.91 |
| 11 | 2.26 | 1.7 |
| 12 | 2.21 | 1.59 |
| 13 | 2.18 | 1.5 |
| 14 | 2.14 | 1.46 |
| 15 | 2.14 | 1.43 |
| 16 | 2.13 | 1.4 |
| 17 | 2.14 | 1.36 |
| 18 | 2.09 | 1.38 |
| 19 | 2.19 | 1.31 |
| 20 | 2.15 | 1.2 |
| 21 | 2.25 | 1.12 |
| 22 | 2.1 | 0.675 |
| 23 | 1.07 | 0.582 |
| **All** | **2.18** | **1.73** |

**Table B2. Toolik model lake temperature errors for inflow and no-inflow simulations split by season.**

| | | Season | | | |
|---|---|---|---|---|---|
| | | Frozen | | Thawed | |
| | Scenario | Inflow | No inflow | Inflow | No inflow |
| Metric | Depth, m | | | | |
| RMSE | 1 | - | - | 1.29 | **1.22** |
| | 3 | 1.64 | **0.877** | 2.69 | **2.88** |
| | 4 | 1.98 | **1.12** | **2.70** | 3.77 |
| | 10 | 1.99 | **0.996** | **2.96** | 3.09 |
| | 19 | 1.99 | **0.931** | 2.56 | **1.89** |
| | All | 1.85 | **0.967** | **2.41** | 2.59 |
| MAE | 1 | - | - | 1.12 | 1.03 |
| | 3 | 1.46 | 0.628 | 1.89 | 2.00 |
| | 4 | 1.73 | 0.865 | 1.87 | 2.81 |
| | 10 | 1.65 | 0.772 | 2.42 | 2.57 |
| | 19 | 1.65 | 0.863 | 2.01 | 1.80 |
| | All | 1.55 | 0.76 | 1.78 | 1.92 |
| Bias | 1 | - | - | -1.01 | -0.999 |
| | 3 | 1.26 | -0.361 | -1.39 | -0.797 |
| | 4 | 1.35 | -0.636 | 0.15 | 2.32 |
| | 10 | 1.13 | -0.54 | 0.742 | 2.31 |
| | 19 | 1.32 | -0.836 | 0.527 | 1.52 |
| | All | 1.21 | -0.563 | -0.235 | 0.725 |

**Appendix C: Modeled water temperature, soil temperature, and heat flux.**

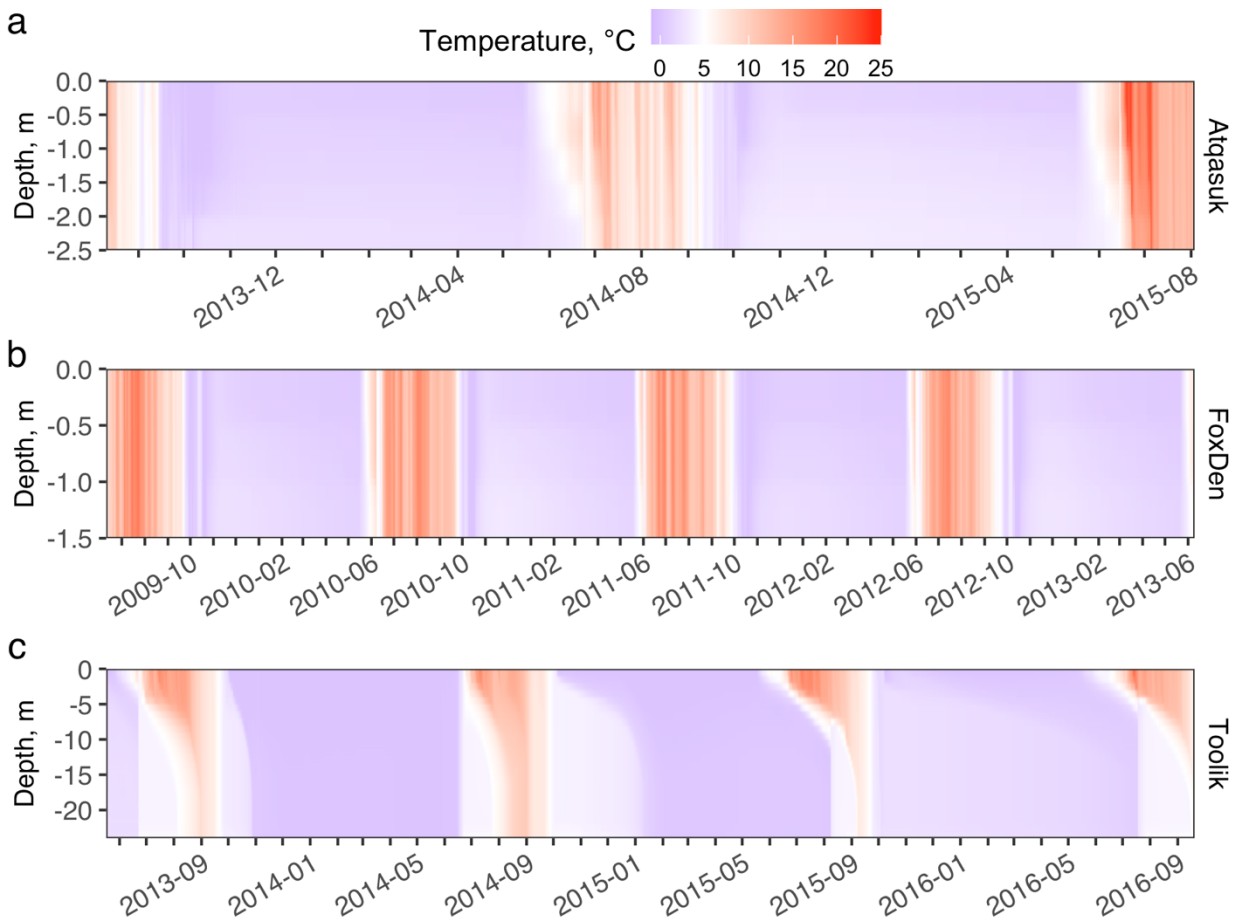

**Figure C1. Baseline lake water temperature over the simulation period for Atqasuk (a), Fox Den (b), and Toolik Lake (c).**


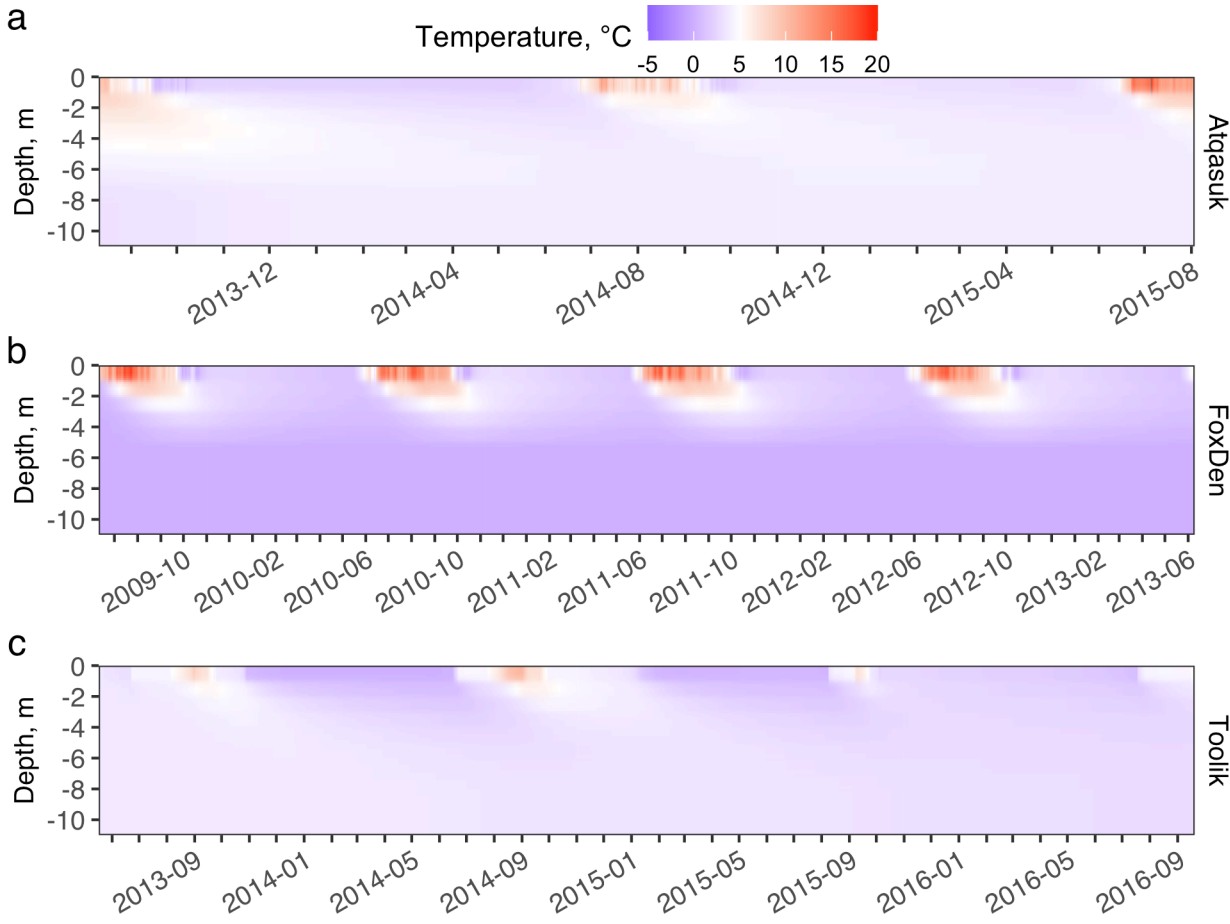

**Figure C2. Baseline lake sediment temperature over the simulation period for Atqasuk (a), Fox Den (b), and Toolik Lake (c) from 0 m to 10 m depth.**

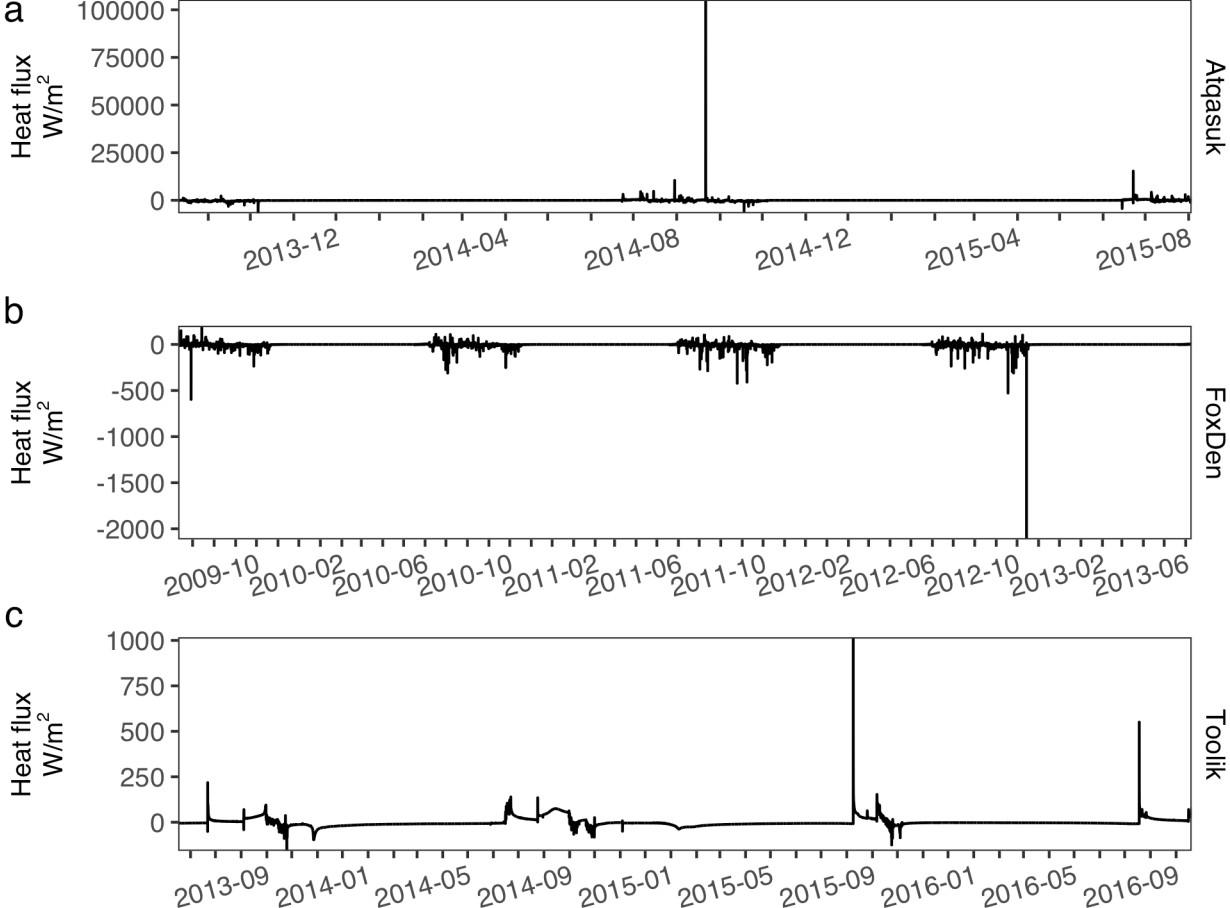


**Figure C3. Downward heat flux at the bottom of the water column over the baseline simulation period for Atqasuk (a), Fox Den (b), and Toolik Lake (c).**

**Appendix D: Water vertical resolution sensitivity analysis**

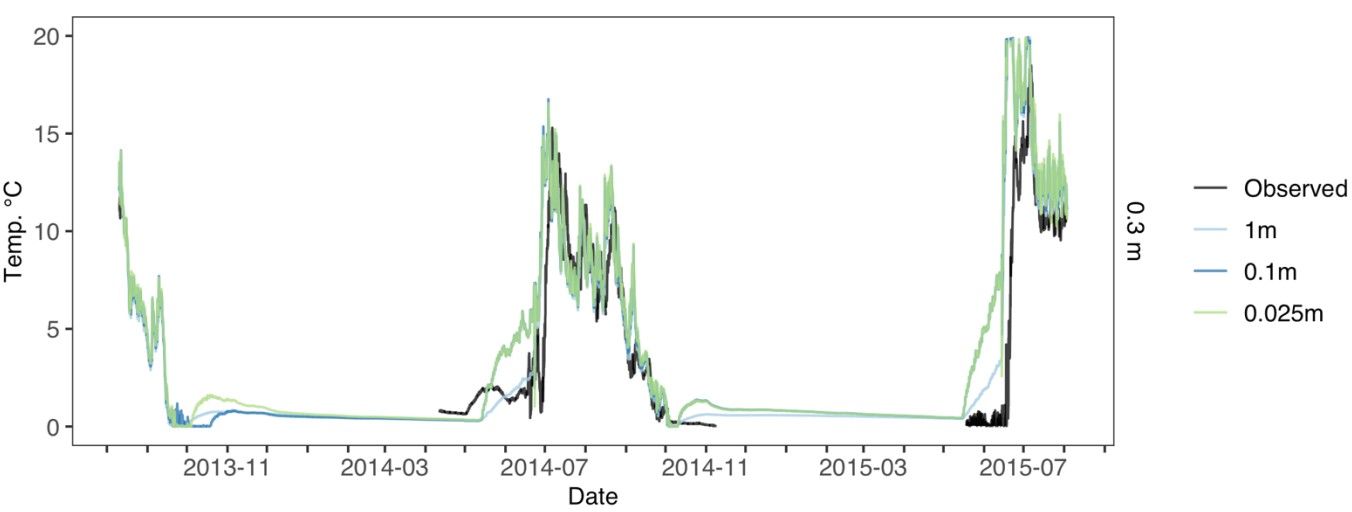


**Figure D1. Atqasuk 0.3m water temperature across different water vertical resolution simulations. Observed water temperature in black, 1m vertical resolution in light blue, 0.1m vertical resolution in blue, 0.025m vertical resolution in green. Note 0.1m and 0.025m overlap for most of the time series.**

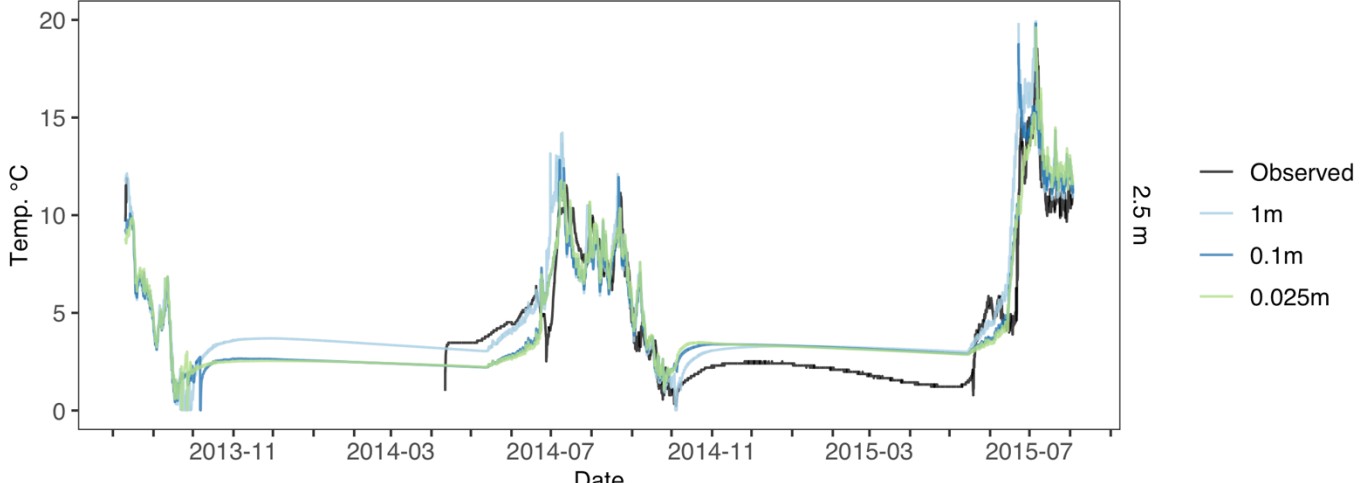

Figure D2. Atqasuk 2.5m water temperature across different water vertical resolution simulations. Observed water temperature in black, 1m vertical resolution in light blue, 0.1m vertical resolution in blue, 0.025m vertical resolution in green. Note 0.1m and 0.025m overlap for most of the time series.

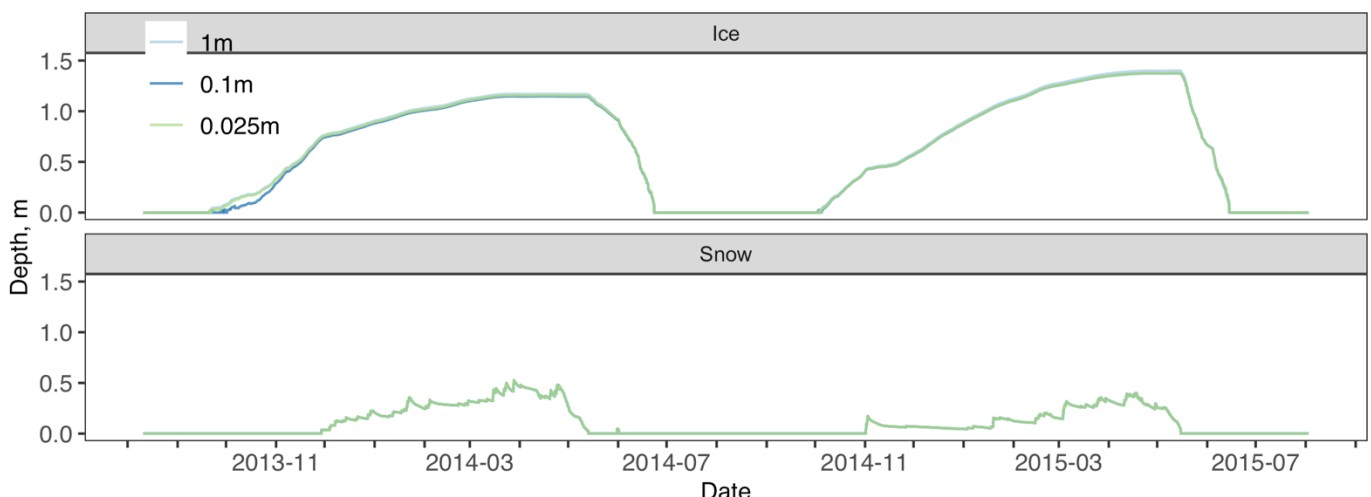

Figure D3. Atqasuk ice and snow depth across different water vertical resolution simulations. No observed data available, 1m vertical resolution in light blue, 0.1m vertical resolution in blue, 0.025m vertical resolution in green. Note data overlaps for most of the time series.

Table D1. Atqasuk modeled water temperature errors calculated from measured water temperature time series at 0.3m and 2.5m across different water vertical resolution simulations. Bold rows are the lowest RMSE for a given water depth. Mean absolute error (MAE), root mean square error (RMSE), mean average error (Bias), percent Bias, mean absolute percent error (MAPE), mean z-score (zScore mean), and median z-score (zScore median).

| Scenario | Depth (m) | Water Res. (m) | MAE | RMSE | Bias |
|---|---|---|---|---|---|
| Atgasuk_w0_025m | 0.3 | 0.025m | 5.07 | 7.16 | -4.77 |
| Atgasuk_w0_065m | | 0.065m | 5.07 | 7.15 | -4.74 |
| Atgasuk_w0_1m | | 0.1m | 5.06 | 7.14 | -4.72 |
| **Atgasuk_w1m** | | **1.0m** | **4.58** | **6.87** | **-4.15** |
| **Atgasuk_w0_025m** | **2.5** | **0.025m** | **1.3** | **1.44** | **-0.481** |
| **Atgasuk_w0_065m** | | **0.065m** | **1.29** | **1.44** | **-0.448** |

| | | | | |
|---|---|---|---|---|
| | | | | - |
| **Atgasuk_w0_1m** | **0.1m** | **1.29** | **1.44** | **0.491** |
| | | | | - |
| Atgasuk_w1m | 1.0m | 1.21 | 1.64 | 0.834 |

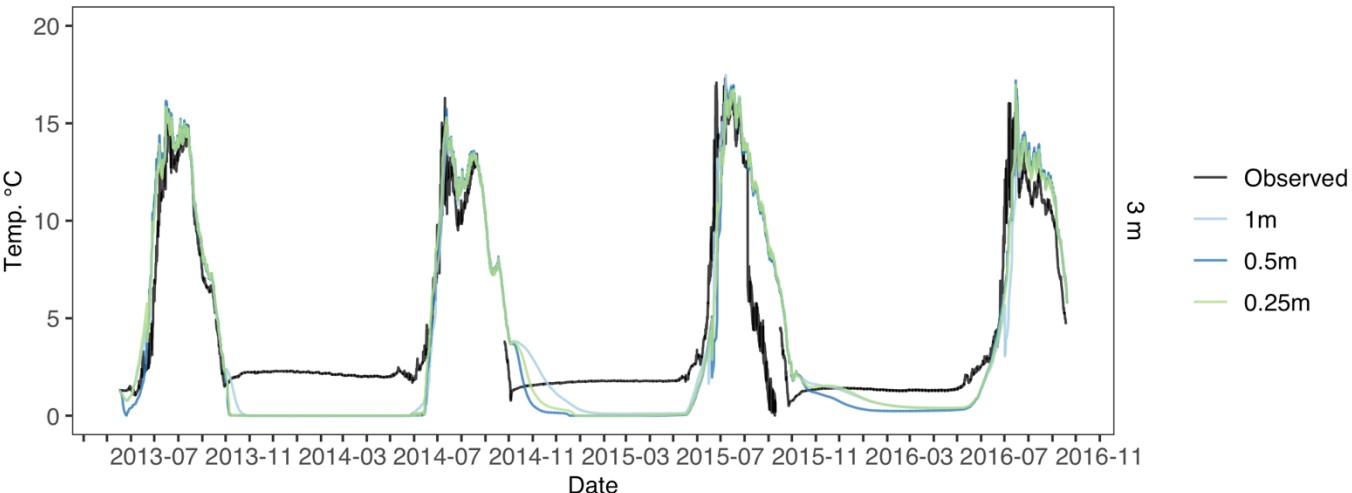

**Figure D4. Toolik 1m water temperature across different water vertical resolution simulations. Observed water temperature in black, 1m vertical resolution in light blue, 0.5m vertical resolution in blue, 0.25m vertical resolution in green.**


**Figure D5. Toolik 3m water temperature across different water vertical resolution simulations. Observed water temperature in black, 1m vertical resolution in light blue, 0.5m vertical resolution in blue, 0.25m vertical resolution in green.**

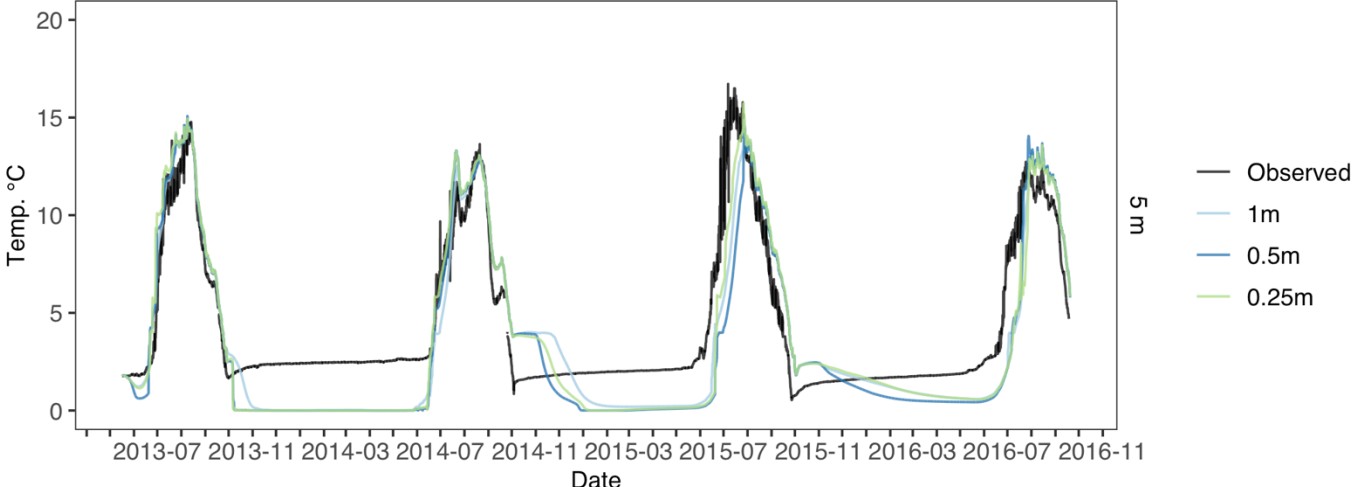


**Figure D6.** Toolik 5m water temperature across different water vertical resolution simulations. Observed water temperature in black, 1m vertical resolution in light blue, 0.5m vertical resolution in blue, 0.25m vertical resolution in green.

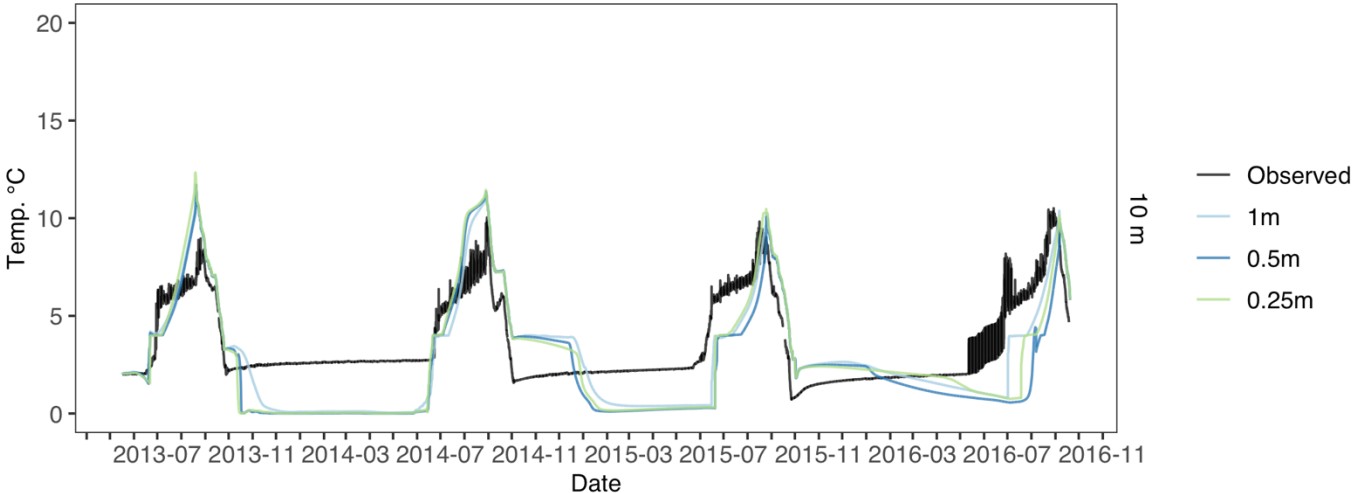

**Figure D7.** Toolik 10m water temperature across different water vertical resolution simulations. Observed water temperature in black,
1m vertical resolution in light blue, 0.5m vertical resolution in blue, 0.25m vertical resolution in green.

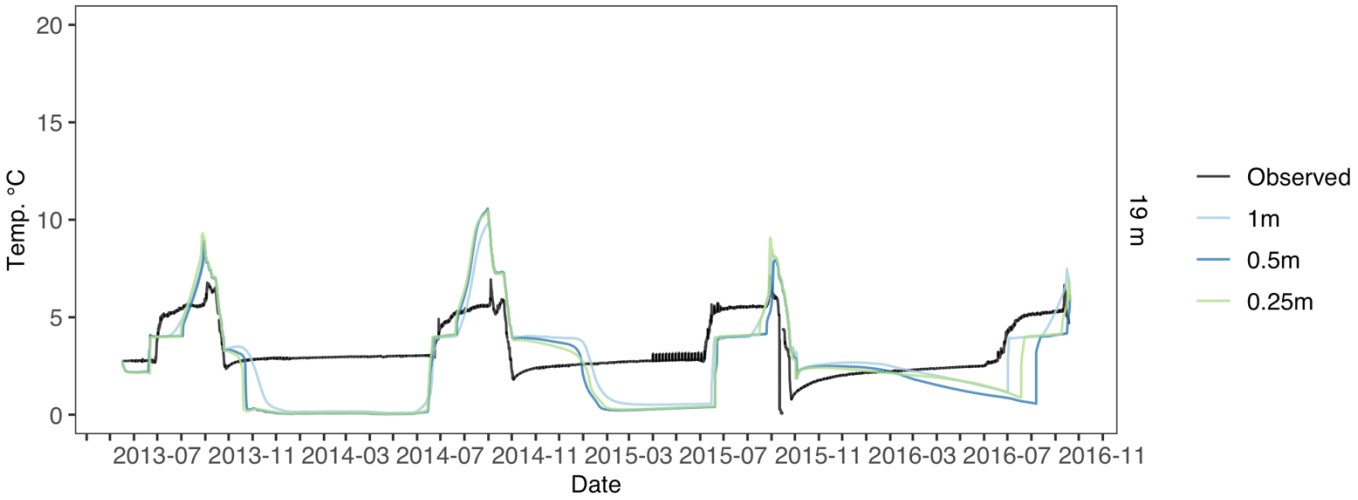

**Figure D8.** Toolik 19m water temperature across different water vertical resolution simulations. Observed water temperature in black, 1m vertical resolution in light blue, 0.5m vertical resolution in blue, 0.25m vertical resolution in green.

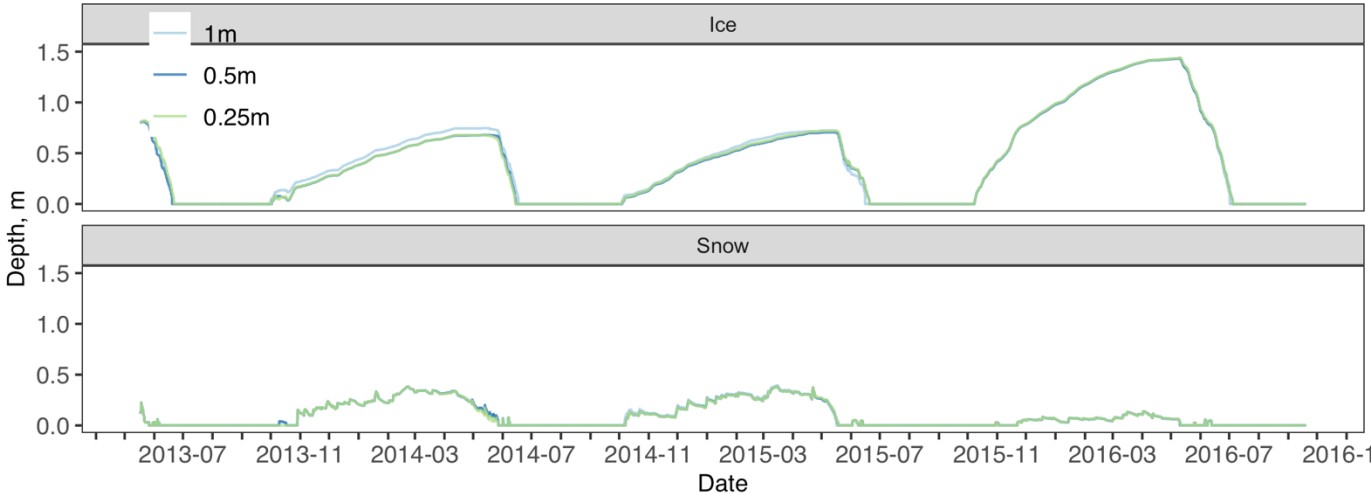

 **Figure D9. Toolik ice and snow depth across different water vertical resolution simulations. No observed data available, 1m vertical resolution in light blue, 0.5m vertical resolution in blue, 0.25m vertical resolution in green.**

**Table D2. Toolik water temperature error calculated from measured water temperatures at multiple depths across different water vertical resolution simulations. Bold rows are the lowest RMSE for a given water depth. Mean absolute error (MAE), root mean square error (RMSE), mean average error (Bias), percent Bias, mean absolute percent error (MAPE), mean z-score (zScore mean), and median**
**z-score (zScore median).**

| Scenario | Depth (m) | Water Res. (m) | MAE | RMSE | Bias |
|---|---|---|---|---|---|
| Too_w0_25m | 1 | 0.25 | 1.19 | 1.42 | -0.905 |
| Too_w0_5m | | 0.5 | 1.18 | 1.36 | -1 |
| **Too_w0_65m** | | **0.65** | **1.12** | **1.29** | **-1.01** |
| Too_w1m | | 1.0 | 1.15 | 1.35 | -0.976 |
| Too_w0_25m | 3 | 0.25 | 1.61 | 2.08 | 0.469 |
| Too_w0_5m | | 0.5 | 1.69 | 2.15 | 0.61 |
| **Too_w0_65m** | | **0.65** | **1.59** | **2.02** | **0.464** |
| Too_w1m | | 1.0 | 1.59 | 2.1 | 0.421 |
| **Too_w0_25m** | **5** | **0.25** | **1.7** | **2.01** | **0.772** |
| Too_w0_5m | | 0.5 | 1.82 | 2.23 | 1 |
| Too_w0_65m | | 0.65 | 1.78 | 2.24 | 0.964 |
| **Too_w1m** | | **1.0** | **1.67** | **2.01** | **0.76** |
| Too_w0_25m | 10 | 0.25 | 1.74 | 2.06 | 0.69 |
| Too_w0_5m | | 0.5 | 1.89 | 2.2 | 0.931 |
| Too_w0_65m | | 0.65 | 1.9 | 2.35 | 1 |
| **Too_w1m** | | **1.0** | **1.6** | **1.84** | **0.57** |
| Too_w0_25m | 19 | 0.25 | 1.63 | 1.98 | 0.808 |
| Too_w0_5m | | 0.5 | 1.76 | 2.09 | 0.951 |
| Too_w0_65m | | 0.65 | 1.77 | 2.19 | 1.07 |
| **Too_w1m** | | **1.0** | **1.49** | **1.78** | **0.594** |

**Appendix E. Soil vertical resolution sensitivity analysis**

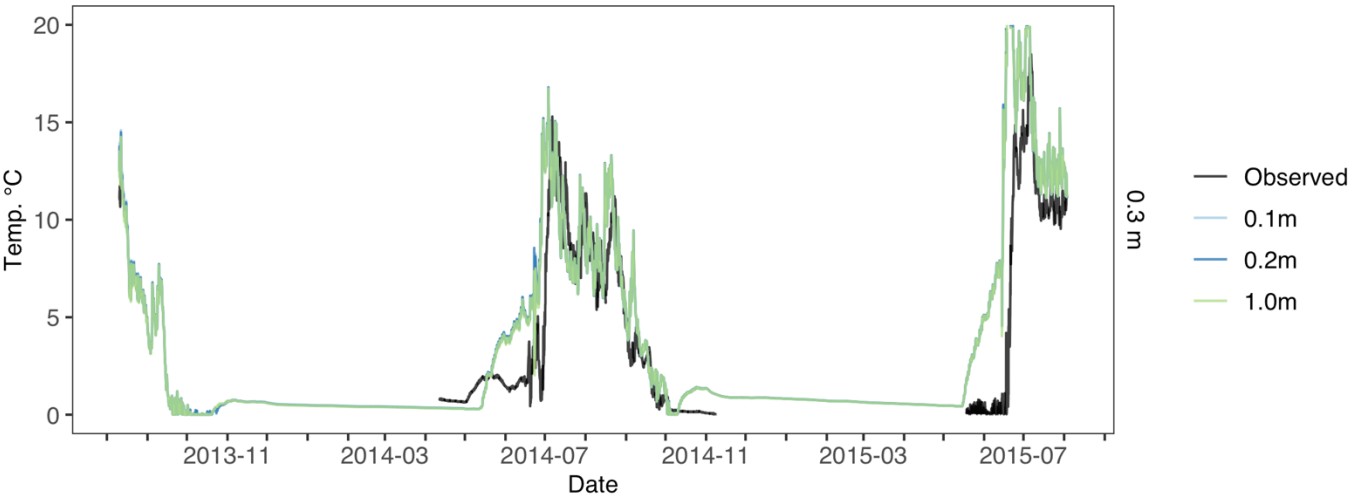

**Figure E1. Atqasuk 0.3m water temperature across different soil vertical resolution simulations. Observed water temperature in black, 0.1m vertical resolution in light blue, 0.2m vertical resolution in blue, 1.0m vertical resolution in green.**

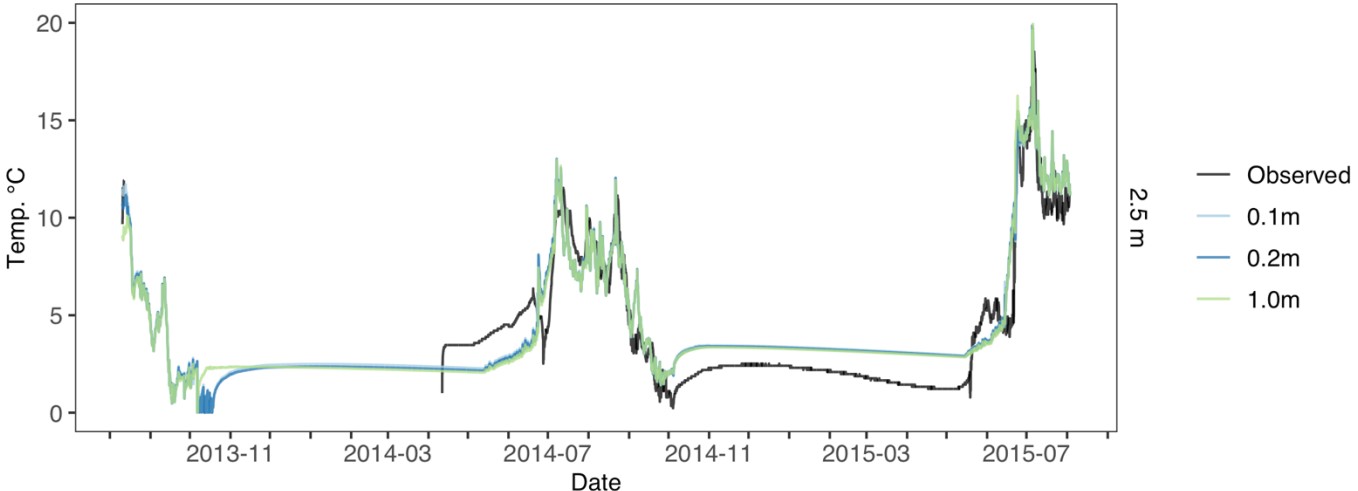

**Figure E2. Atqasuk 2.5m water temperature across different soil vertical resolution simulations. Observed water temperature in black, 0.1m vertical resolution in light blue, 0.2m vertical resolution in blue, 1.0m vertical resolution in green.**

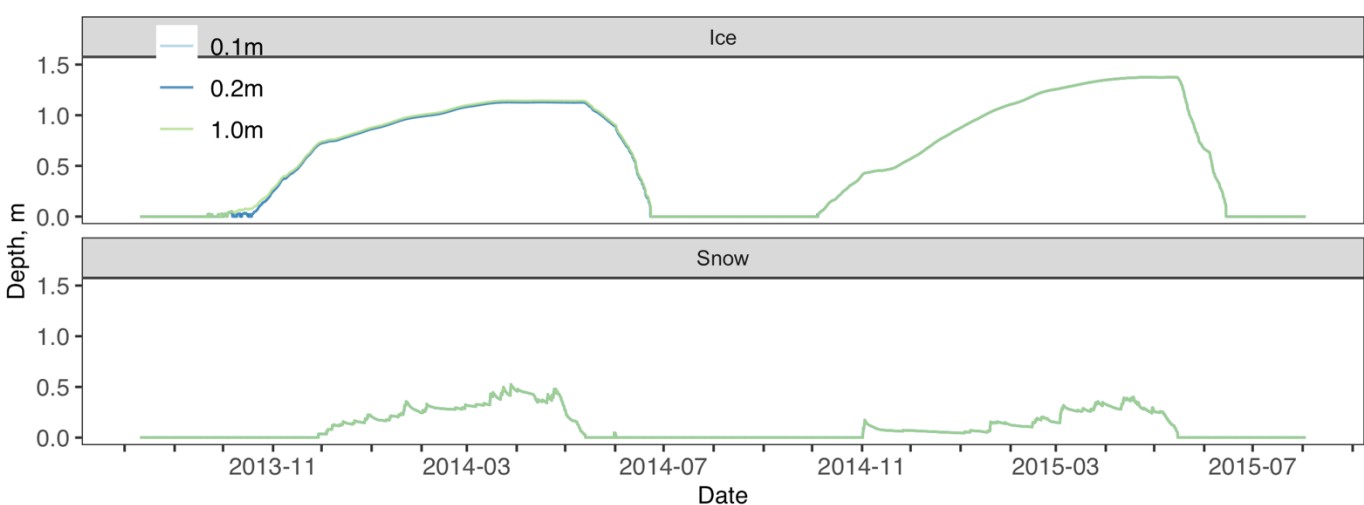


**Figure E3. Atqasuk ice and snow depth across different soil vertical resolution simulations. No observed data available, 0.1m vertical resolution in light blue, 0.2m vertical resolution in blue, 1.0m vertical resolution in green.**

**Table E1. Atqasuk modeled water temperature errors calculated from measured water temperature time series at 0.3m and 2.5m across different soil vertical resolution simulations. Bold rows are the lowest RMSE for a given water depth. Mean absolute error (MAE), root mean square error (RMSE), mean average error (Bias), percent Bias, mean absolute percent error (MAPE), mean z-score (zScore mean), and median z-score (zScore median).**

| Scenario | Depth (m) | Soil Res. (m) | MAE | RMSE | Bias |
|---|---|---|---|---|---|
| Atgasuk_s0_1m | 0.3 | 0.1 | 5.09 | 7.17 | -4.76 |
| Atgasuk_s0_5m | | 0.5 | 5.08 | 7.16 | -4.76 |
| **Atgasuk_s1m** | | **1.0** | **5.07** | **7.15** | **-4.74** |
| **Atgasuk_s0_1m** | **2.5** | **0.1** | **1.3** | **1.44** | **- 0.541** |
| **Atgasuk_s0_5m** | | **0.5** | **1.29** | **1.44** | **-0.5** |
| **Atgasuk_s1m** | | **1.0** | **1.3** | **1.44** | **- 0.448** |

## Appendix F. Temporal resolution sensitivity analysis

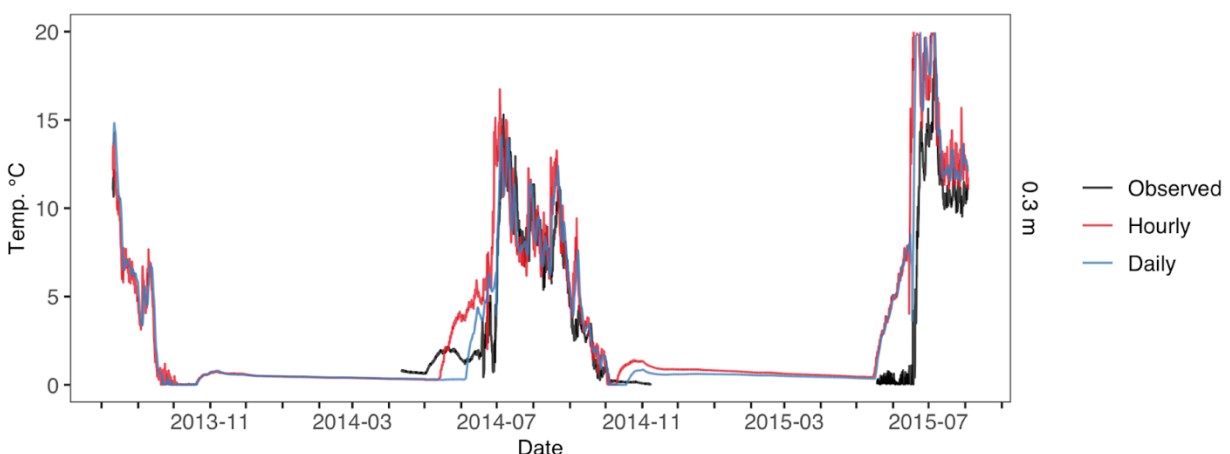

**Figure F1. Atqasuk 0.3m water temperature across different temporal resolution simulations. Observed water temperature in black, hourly temporal resolution in red and daily temporal resolution in blue.**

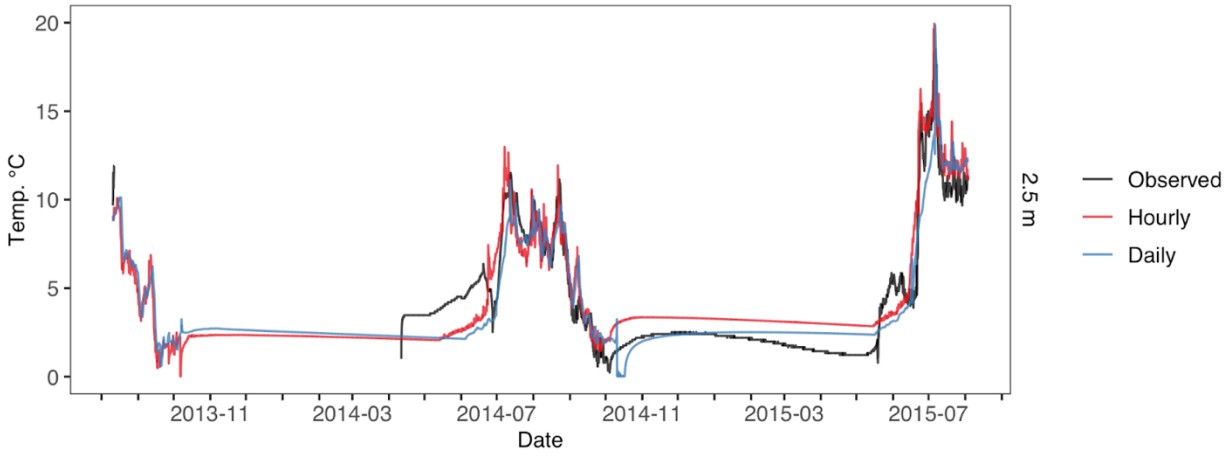

**Figure F2. Atqasuk 2.5m water temperature across different temporal resolution simulations. Observed water temperature in black, hourly temporal resolution in red and daily temporal resolution in blue.**

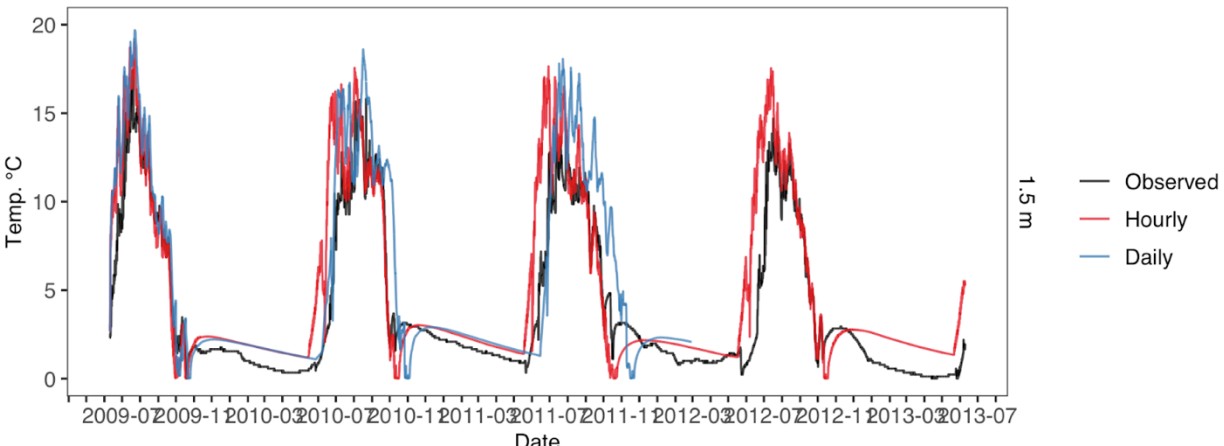

**Figure F3. Fox Den 1.5m water temperature across different temporal resolution simulations. Observed water temperature in black, hourly temporal resolution in red and daily temporal resolution in blue.**

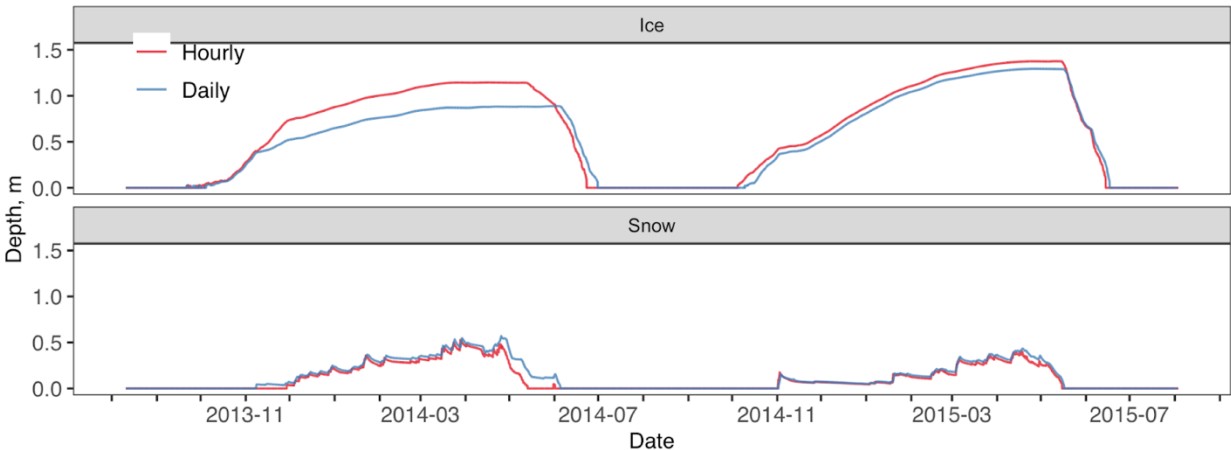


**Figure F4. Atqasuk ice and snow depth across different temporal resolution simulations. No observed data available, hourly temporal resolution in red and daily temporal resolution in blue.**

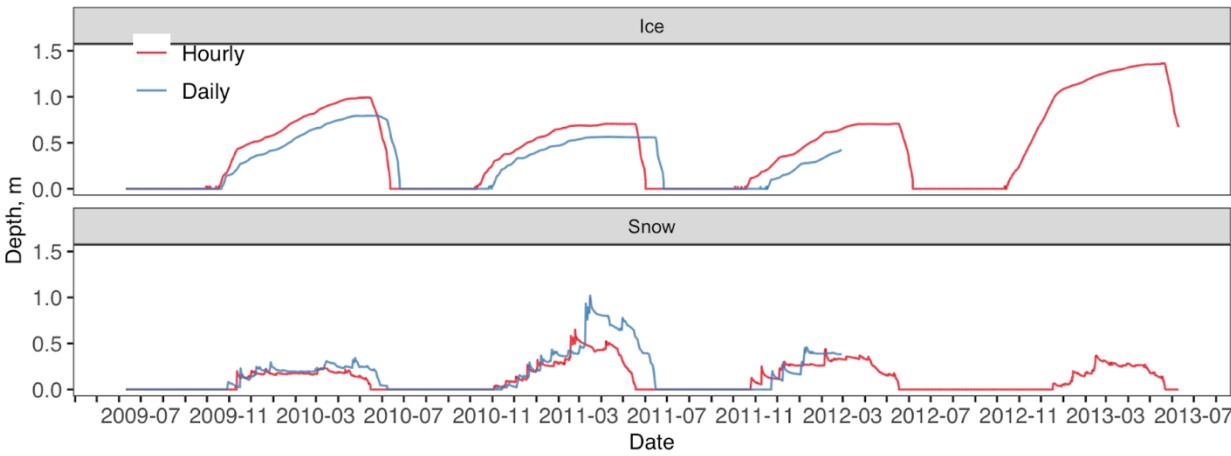

**Figure F5. Fox Den ice and snow depth across different temporal resolution simulations. No observed data available, hourly temporal**
**resolution in red and daily temporal resolution in blue.**

**Table F1. Atqasuk modeled water temperature errors calculated from measured water temperature time series at 0.3m and 2.5m across different temporal resolution simulations. Bold rows are the lowest RMSE for a given water depth. Mean absolute error (MAE), root mean square error (RMSE), mean average error (Bias), percent Bias, mean absolute percent error (MAPE), mean z-score (zScore mean), and median z-score (zScore median).**

| Site | Temporal Res. | Depth, m | MAE | RMSE | Bias |
|---|---|---|---|---|---|
| Atqasuk | Hourly | 0.3 | 5.07 | 7.15 | -4.74 |
| | **Daily** | | **4.7** | **6.83** | **-4.33** |
| | Hourly | 2.5 | 1.29 | 1.44 | -0.448 |
| | **Daily** | | **1.02** | **1.31** | **0.24** |
| Fox Den | Hourly | 1.5 | 1.69 | 2.47 | -1.27 |
| | **Daily** | **1.5** | **1.45** | **2.41** | **-0.953** |


## 7 Code and data availability

The atmospheric forcing data and model input files can be found at (Jafarov et al., 2021) or upon request from the corresponding author. The model outputs, validation datasets, and processing scripts can be found at (Clark and Jafarov, 2021) or upon request from the corresponding author. The source code for the latest
version of the LAKE 2.0 model is available at (http://tesla.parallel.ru/Viktor/LAKE/wikis/LAKE-model, Stepanenko et al., 2016).

## 8 Author contributions

JC and EJ were responsible for the study design, numerical simulation setup, model runs, processing and analysis of the results. JC and EJ drafted the manuscript. JC, EJ, VS, KT, and BJ edited and revised the
manuscript.

## 9 Competing interests

The authors declare that they have no conflict of interest.

## 10 Acknowledgements

JAC and KDT acknowledge NSF Award #1850578. EEJ received support as part of the Interdisciplinary
Research for Arctic Coastal Environments (InteRFACE) project through the Department of Energy, Office of Science, Biological and Environmental Research (BER) Regional and Global Model Analysis (RGMA) program, awarded under contract grant 89233218CNA000001 to Triad National Security, LLC ("Triad"). VS acknowledges support from the Russian Ministry of Science and Higher Education (grant MD-1850.2020.5, and agreement 075-15-2019-1621). BMJ was supported by grants from the US National
Science Foundation under award #'s OPP-1850578, OPP-2114051, and OPP-1806213.

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
