# Peer review of "Thermal modeling of three lakes within the continuous permafrost zone in Alaska using LAKE 2.0 model"

_Geoscientific Model Development, 2022_

## Author Response (AR1)

GMD responses to Reviewer comments.
Reviewer comments in grey, responses in black.

We thank both reviewers for their effort and insightful comments. These are two of the more keen and constructive reviews that we have received. Reviewers identified several key issues: model validation and performance, model temporal resolution, model vertical resolution, sediment temperatures, and Toolik lake inflow data. We addressed these issues by adding model evaluation metrics, adding new appendices for model resolution sensitivity, correcting errors, and updating manuscript text to address comments. We provide detailed responses to these issues below. As the GMD interactive comments do not allow us to submit a revised manuscript at this stage, we have attached excerpts from our revised manuscript that should be viewed along with our responses.

**Major comments:**

Comment: "I have some issues with the study. Firstly, I think that the model baseline simulations were not correctly validated. I can´t fully evaluate the model performance, and/or compare the model results with other model simulations (e.g. Guo et al., 2021, modeled Toolik lake) without a model evaluation metric such as: mean absolute error (MAE) or root mean square error (RMSE). Furthermore, I don´t understand how the model was calibrated. What function were you trying to minimize in order to optimize the model performance?"

Response: Thank you for your effort and insightful comments. We have updated the manuscript with model evaluation metrics (including MAE and RMSE) as requested. Model performance was similar to Guo et al. 2021, with RMSE ~2C. However, it should be noted Guo et al. 2021 simulated Toolik lake only for the thawed seasons of 1983-1988. The LAKE model was minimally calibrated for each lake, as described in Section 2.2, to initialize water and sediment temperatures. A standard set of model parameters were applied to all lakes to demonstrate the applicability of the LAKE model in simulating Arctic lakes (Table 1).

Comment: "Secondly, why didn´t you show the lakes sediment temperature obtained with the model as a function of water temperature? This kind of data is quite relevant for other researchers."

Response: We have added results showing sediment temperatures and new figures for water temperature profiles in addition the figures already present showing water temperatures (Appendix C). As the focus of this paper was not directly on lake sediment temperatures we did not attempt to demonstrate sediment temperatures as a function of water temperatures.

**Specific comments:**

Comment: "L25: I think that the word "completes" is very strong."

Response: We changed it to 'is'.

Comment: "L26-L29: This sentence is unclear to me. You say that the model "is not highly sensitive to the weather data perturbations", and you conclude that "snow depth and lake ice strongly affect water temperatures during the frozen season"?"

Response: We have updated the text to clarify our point. "The sensitivity analysis shows us that lake water temperature is not highly sensitive to small changes in air temperature or precipitation, while changes in shortwave radiation and large changes in precipitation produced larger effects. Snow depth and lake ice strongly affect water temperatures during the frozen season which dominates the annual thermal regime. These findings suggest that reductions in lake ice thickness and duration could lead to more heat storage by lakes and enhanced permafrost degradation."

Comment: "L31: I suggest the following change to this sentence: "Approximately forty percent…""

Response: We have made this change.

Comment: "L70: **Description of the model:** I think that you need to improve the model description, namely, the multilayer snow and ice modules (Stepanenko and Lykossov, 2005; Stepanenko et al., 2011)."

Response: We have elaborated on this section to include a description of the snow and ice modules, including references.

Comment: "L85: **LAKE model setup:** Please describe the calibration procedure. Which parameters were calibrated in which ranges? Was calibration automatic? Please describe the parameters of the baseline simulation. The table 1 included in Stepanenko et al. (2016) is a very good example."

Response: Our calibration procedure simply involved the initialization of the soil and water temperature values as described in the Section 2.2. No other parameters were calibrated. The parameters of the baseline scenarios have been added as Table 1.

Comment: "L94: **Input data**: Please describe all meteorological variables. How did you characterize the inflow water temperature to lake Toolik? Please describe the initial water temperature and sediments values, before and after the 10 years simulation."

Response: We have added text to section 2.3 describing all met variables. Inflow water temperature was measured daily with discharge. Water temperature is included in the inflow input file. Discharge and temperature are described in section 2.7. Initial water temperature was taken from observed water temperature data. Initial and calibrated sediment temperatures are now reported in Table 1.

Comment: "L140: Please replace $Wm^{-1}$ with $Wm^{-2}$."

Response: Thanks, we made this change.

Comment: "L150: Do you have lake water level values? Do you think that neglecting the lake water level may lead to errors in surface heat flux predictions?"

Response: Interesting point. We do not have observations of lake water level values. The water level change may affect surface fluxes via the thickness of the mixed (or active) layer of a lake, the latter is a layer which total heat capacity interacts with the atmosphere. If not limited by lake depth, the typical summertime ML thickness in mid- and high latitudes is 3-5 m (see e.g. simulated/observed temperature profiles in LakeMIP papers). Thus, there are two situations with respect to the lake level effects on ML depth and thus the surface fluxes. First, the lake is shallower than 3-5 m, then the ML is a lake depth. In this case, the water level may affect fluxes, if it varies significantly retaining the depth below 3-5 m. In the case where the lake depth much exceeds 3-5 m (Toolik lake), the level variations do not change ML depth and thus the fluxes.

Comment: "L156: I suggest adding a new section, "Evaluation metrics" for the "new" evaluation metrics (e.g. RMSE). The Z-score equation can also be included here.  You don't need to apply the "new" metrics to the sensitivity analysis."

Response: We have added this section, now section 2.8.

Comment: "L169: "During the frozen season, the modeled temperatures underestimate cooling in the lake." By how much?"

Response: We have added Table 2 which shows model error (MAE, RMSE, Bias) for the entire time series, and split by frozen and thawed season. For this particular sentence the error for Atqasuk over the frozen period was 5.8 (RMSE).

Comment: "L189-190: "For 2013 and 2014 the modeled shallow (0, 3 m) water temperature was overestimated while for 2015 and 2016 shallow water temperature was underestimated, though it tracked observed temperature." By how much?"

Response: We have added Table 2 which shows model error (MAE, RMSE, Bias) for the entire time series, and split by frozen and thawed season. The Toolik model simulations have been updated based on corrected discharge data. This sentence and interpretation of the Toolik water temperatures have been changed. Thawed and frozen season errors are presented in Table 2.

Comment: "L192: I can´t see the step-like dip in figures B1 and B2 can this fact be related with inflow water temperature?"

Response: We thank the keen reviewer who caught this error.  We were able to trace the 'dips' to a formatting error in the inflow data file.  This has been corrected. All Toolik simulations have been repeated and figures updated (Section 3.3).  The 'dips' were an artifact of the erroneous inflow data and are no longer present (Figs. 3 & 4).

Comment: "L200: The datasets length (x values) shown in figures 3 and 4 is smaller than the datasets length shown in figures B1 and B2."

Response: These have been corrected to show the same length of data.

Comment: "L210: "shallow depth water temperatures (1, 3, and 5 m 210 depth, -0.13 to 0.34)". I can´t find the value -0.13 in Figure 5."

Response: This was an error. The text has been updated to reflect the data in the figure. Please note this figure and data have been updated to reflect the new simulations for Fox Den (now hourly) and Toolik (with corrected inflow data)(Fig. 5).

Comment: "L246: "Modeled shallow water (1 m) temperature exceeded the observed temperatures" After the incorporation of inflows/outflows, the water temperature (1 m) in 2013 and 2014, still exceeds observed water temperatures. This kind of analysis would be easier with a model evaluation metric."

Response: Error metrics have been added and are included in Table 2, B1, & B2 for this sentence.

Comment: "L270: I think that this entire section "Modeling Lake thermal effects in permafrost" must be in the introduction."

Response: We have moved this section to the Introduction.

Comment: "L286: "The "dips" of water temperature in LAKE model results for Toolik lake down to depths of 10 m prior to ice-off can be explained". I can see the dip at 19 m (Figure 4, 2014-07)."

Response: We thank the keen reviewer who caught this error.  We were able to trace the 'dips' to a formatting error in the inflow data file.  This has been corrected. All Toolik simulations have been repeated and figures updated (Section 3.3).  The 'dips' were an artifact of the erroneous inflow data and are no longer present (Figs. 3 & 4).

Comment: "L287: "can be explained by convective instability under the ice, where this instability can be caused by the under-ice penetration of solar radiation" As I said previously, I can´t see the "dips" in figures B1 and B2. Can this be related with the effect of lake inflow?"

Response: We thank the keen reviewer who caught this error.  We were able to trace the 'dips' to a formatting error in the inflow data file.  This has been corrected. All Toolik simulations have been repeated and figures updated (Section 3.3).  The 'dips' were an artifact of the erroneous inflow data and are no longer present (Figs. 3 & 4).

R2:
Comment: "Modeling of lake thermodynamics in polar regions is a highly relevant topic with regard to the response of the Arctic permafrost to the global change. The model LAKE has been intensively applied in recent studies on lake dynamics and air-lake interaction. Therefore, a study on the LAKE model abilities to simulate thermal properties of lakes in the permafrost zone falls into the scope of the GMD and is of interest for its readership. Comparison of the model performance for three Arctic lakes of different morphometry provides a necessary background for future analysis of the atmosphere-lake-permafrost interaction. Herewith, the study is a valuable contribution to modeling of lakes as components of the climate system. The methods, presentation of results, and discussion are generally adequate to the problem statement, however contain some gaps, related, in particular, to the effects of the spatial and temporal resolution on the modeling results and to the simulation of the water-sediment interaction as a crucial aspect of lake modeling in the permafrost zone. I recommend extending the study with relevant details providing the reader with a necessary overview of the model performance beyond the sensitivity to variations in the input forcing, which is currently the major focus of the manuscript."

Response: Thank you for your comments. We have added several new sections to the manuscript and to the Appendix that we believe add more detail and aid in understanding the spatial resolution an temporal resolution on modeling results as well as results of the water-sediment interaction.

Comment: "As it was pointed out by the previous reviewer, the model validation is presented in a rather qualitative way, and some numerical scores of the model performance, like bias, absolute error, RMSE etc., will be useful here."

Response: We have updated the manuscript with model evaluation metrics (including MAE and RMSE) as requested. Please see Table 2 and the Appendix.

Comment: "The temporal resolution of the model input was different for three different lakes: 1 day for one of them and 1 hour for the two others. It is unclear how the diurnal cycle of the atmospheric forcing and radiation was treated in the model. Were the daily data interpolated on sub-diurnal scales? If yes, how the interpolation was performed? How does the neglect of the sub-daily variations in the input data affect the model output? The question could be answered by comparison of model runs with daily and hourly inputs for the lakes where sub-diurnal data on forcing are available."

Response: We have updated the simulations to use the same temporal resolution (1 hour) for all lakes. Additionally we have added a section to the Appendix that shows the effect of temporal resolution on model performance (Appendix F). Daily data are linearly interpolated to finer temporal scales within LAKE.

Comment: "The vertical resolution for both water column and sediment was set to 1 meter and did not vary between lakes. What were the criteria for the choice of the resolution? One can assume that for the vertical diffusion rates within the sediment of $10^{-6}$ m^2 s^{-1}, the vertical resolution of 1 m will capture the processes with typical time scales of >10 days. Is it sufficient? How many vertical grid points did Fox Den have, whose depth

is 1.5 m? Can you perform sensitivity runs demonstrating the effect of the vertical resolution on the model output?"

Response: We regret that the text incorrectly stated the vertical resolution for the water column. We have corrected the text. We used 40 layers for the water layer for all lakes which results in a different vertical resolution for each lake (Atqasuk=0.065m, Fox Den=0.04m, and Toolik=0.65m, see Table 1). Our experience has shown that 40 layers is sufficient (Stepanenko et al. 2010, 2013, & 2016). We have performed the additional sensitivity runs demonstrating the effect of vertical resolution on model performance and have included the results in Appendix D.

Comment: "L316, Section 5.4 The details on the sediment layer modeling results are crucial for discussion on the model applicability to permafrost lakes. The information is missing in the ms. How did the soil temperatures under the lake bottom vary during the modeling period? What are the values of the bottom heat flux and how do they depend on the model configuration, initial and boundary conditions?"

Response: We have added new figures and a new section in the results to show the sediment temperatures and heat flux during simulations (Appendix C). Soil temperatures responded differently in each lake. In general, shallow sediment showed warming in the thaw period and deeper sediments remained constant over the simulation period.

Some minor remarks:

Comment: " "It is a large lake (2,732,050 m2 )..." why 2 km^2 area is large for a lake?"

Response: It is large relative to our study lakes. We have updated the text reflect this comparison.

Comment: " " 30 cm and 250 cm" better use meters here for consistency."

Response: We have made this change.

Comment: "In Fox Den the model calculated up to 1.0 m thick ice cover in a 1.5 m deep lake. Was the water volume/depth adjusted during the ice-covered period? Was 1 m vertical resolution sufficient for simulation?"

Response: All frozen water (which formed the ice layer) is subtracted from the lake water volume. The water depth is adjusted accordingly. As to resolution, the grid spacing in water and ice is automatically adjusted in the model to keep the predefined number of numerical layers in each physical layer. In the manuscript you reviewed, we misstated the vertical resolution used for the simulations (Table 1). We have corrected these errors. For Fox Den the vertical resolution was 0.0375m which we believe was sufficient for simulation.

Comment: "L286: "The "dips" of water temperature in the LAKE model results for Toolik lake..." How did the vertical model resolution affect the representation of free

convection?  The 1 m resolution seems to be crude for the typical values of the convective layer entrainment rates of < 1 m/day (e.g. Kirillin et al. 2012)."

Response: The statement of 1m resolution was incorrect.  We have corrected the text to reflect the vertical resolution used in each Lake (Table 1).  For Toolik the resolution was 0.65m. We have added an appendix to look at the effects of increasing model water vertical resolution.  Using 1m, 0.5m, and 0.25m vertical resolutions we found minimal effects on lake water temperatures and model performance. Kirillin et al. 2012 report rates of 0.5 m per day increasing to several meters per day in deep lakes. We simulated lakes with vertical resolutions of 0.0635m, 0.0375m, and 0.65m (for Atqasuk, Fox Den, and Toolik respectively) and tested vertical resolutions down to 0.25m for Toolik and 0.025m for Atqasuk (Appendix D). We did not see evidence that the vertical resolutions used in the manuscript was too coarse.

---

## Referee Report (RR1)

The authors fulfill important research for Arctic limnology. By assessing the performance of the LAKE model in simulating lake water temperatures across lakes, seasons, depths and soil and water column resolutions, they make a strong case that LAKE is suitable for future studies of lake-permafrost interactions. In view of the manuscript overview, it is also clear that the authors made strong additions to the paper in response to earlier referee comments. With this in mind, I find this paper to be acceptable for publication at GMD, subject to what are mostly minor revisions and small details that I recommend addressing below.

1) This paper is framed as a necessary step toward understanding lake-permafrost interactions, a very important topic to many geoscience fields. Yet, I find the conclusion misses what future developments will be necessary to apply the model to this end. E.g., in each lake's description in the methods, it is repeated that underlying permafrost presence and/or depths are not yet known. Would bridging such an observational shortcoming be necessary for future applications of LAKE to more directly model lake-permafrost interactions? What other limitations must be overcome for this type of research to move forward? I feel the research progression for which this study's validation is necessary could be better wrapped up.

2) If this is within the bounds of figure/page limits, I think it would be valuable for readers if you present the study sites by adding a map of Alaska, or some explicit spatial domain, that shows the locations of these three lakes.

3) Is there an explanation for why only one lake's (Atqasuk) sensitivity to soil column resolution was tested and the others were not? I note that a lake of median depth among the study sites was chosen, but I don't know if we might expect more or less sensitivity in the other lakes. Perhaps less sensitivity for the deepest lake and more for the shallowest lake? I don't need to see further simulations added to this end, but perhaps some explanation could be provided to justify the single lake approach for this part of the analysis.

Minor details:

L70-72: More correct to explicitly state "in Arctic lakes" at some point here.

L82: "similar types of models" instead of "similar type models".

L93: acronym "MAAT" unnecessary since it only appears once.

L106: "k-eps" should be k-$\varepsilon$?

L103: units should be [$Wm^{-2}$] instead of [$Wm^{-1}$].

L214: "Scenario data" was a bit confusing upon first read. Suggest adding "Scenario data used in the model sensitivity analysis (section ref) was compared ..."

L255: Table 2 should explicitly state that performance metrics are for water temperatures, i.e. "LAKE model performance for Atqasuk, Fox Den, and Toolijk Lake water temperatures..."

L310: Referencing Figure E1 and Table E1, which also have captions that should better directly state what the figure displays water temperature simulations as a function of. Currently, the section header of Appendix E describes that this is a soil column resolution sensitivity analysis. However, figures should be readable in a stand-alone manner by their captions (and therefore also include

some mention of simulations differing by soil column resolution). I recommend scanning the other figures/tables for this property.

L346: Add a space between "Toolik Lake" and (Fig. 6)

L376: Section header using an unofficial "met" acronym for meteorological. I would replace for the full word or introduce an acronym and use more widely.

L386: "…as high winds can and do remove…" should be "…as high winds can remove…"

L402: "…and in a fully coupled lake-atmosphere system model the similar sensitivity experiments should provide different estimates." Should be, "…and in a fully coupled lake-atmosphere system, similar sensitivity experiments should provide different estimates"

General: Note that figure references sometimes switch styles from "Figure X" to "Fig. X", I guess this should be consistent.

---

## Author Response (AR2)

GMD-2022-9
Jason A. Clark, Elchin E. Jafarov, Ken D. Tape, Benjamin M. Jones, and Victor Stepanenko
Response to Reviewers #2
Responses are in blue.

Thank you for your addition comments and consideration of our manuscript. We appreciate the time and effort you have given to improving our manuscript.

**Report #1**
Line 345: "…for the shallow lakes (Atqasuk, Fox Den), but for the thawed season for Toolik Lake(Fig. 6)." Please replace "but" with "and".
We have rephrased this sentence to be more clear. Replacing but with and would have changed the intended meaning of the sentence.

This updated paper is well organized with logical sequence. I'm also satisfied with the response from the authors. All the comments have been addressed carefully. The authors have done a good work.

**Report #2**
The authors fulfill important research for Arctic limnology. By assessing the performance of the LAKE model in simulating lake water temperatures across lakes, seasons, depths and soil and water column resolutions, they make a strong case that LAKE is suitable for future studies of lake permafrost interactions. In view of the manuscript overview, it is also clear that the authors made strong additions to the paper in response to earlier referee comments. With this in mind, I find this paper to be acceptable for publication at GMD, subject to what are mostly minor revisions and small details that I recommend addressing below.
1) This paper is framed as a necessary step toward understanding lake-permafrost interactions, a very important topic to many geoscience fields. Yet, I find the conclusion misses what future developments will be necessary to apply the model to this end. E.g., in each lake's description in the methods, it is repeated that underlying permafrost presence and/or depths are not yet known. Would bridging such an observational shortcoming be necessary for future applications of LAKE to more directly model lake-permafrost interactions? What other limitations must be overcome for this type of research to move forward? I feel the research progression for which this study's validation is necessary could be better wrapped up.
Thank you for this insightful comments. We have revised the discussion to clearly state this observational shortcoming and need in the progression of this research.

2) If this is within the bounds of figure/page limits, I think it would be valuable for readers if you present the study sites by adding a map of Alaska, or some explicit spatial domain, that shows the locations of these three lakes.
Maps of the study site locations are available at the respective data archives and have been published previously (Hinkle 2012, Jones et al. 2021, MacIntyre and Cortes 2017). We present the study lake coordinates for the reader to reference both in the text and in Table 1.

3) Is there an explanation for why only one lake's (Atqasuk) sensitivity to soil column resolution was tested and the others were not? I note that a lake of median depth among the study sites was

chosen, but I don't know if we might expect more or less sensitivity in the other lakes. Perhaps less sensitivity for the deepest lake and more for the shallowest lake? I don't need to see further simulations added to this end, but perhaps some explanation could be provided to justify the single lake approach for this part of the analysis.

We have provided a sensitivity analysis for soil column resolution for a single lake as it was requested in the previous round of revisions. For this analysis we compared the effect of soil resolutions on modeled water temperature error. We did not analyze the sensitivity of soil temperature to lake depth. This a valid question that LAKE 2.0 could be used to address, but the question would require an additional analysis based on theoretic lakes with different depths. Observations of lake sediment temperatures would also be valuable for this additional analysis.

Minor details:

L70-72: More correct to explicitly state "in Arctic lakes" at some point here.

We have made this change.

L82: "similar types of models" instead of "similar type models".

We have made this change.

L93: acronym "MAAT" unnecessary since it only appears once.

We have made this change.

L106: "k-eps" should be k-ε?

We have made this change.

L103: units should be [Wm-2 ] instead of [Wm-1 ].

We have made this change.

L214: "Scenario data" was a bit confusing upon first read. Suggest adding "Scenario data used in the model sensitivity analysis (section ref) was compared …"

We have made this change.

L255: Table 2 should explicitly state that performance metrics are for water temperatures, i.e. "LAKE model performance for Atqasuk, Fox Den, and Toolijk Lake water temperatures…"

We have made this change.

L310: Referencing Figure E1 and Table E1, which also have captions that should better directly state what the figure displays water temperature simulations as a function of. Currently, the section header of Appendix E describes that this is a soil column resolution sensitivity analysis. However, figures should be readable in a stand-alone manner by their captions (and therefore also include some mention of simulations differing by soil column resolution). I recommend scanning the other figures/tables for this property.

We have updated the Appendix table and figure captions.

L346: Add a space between "Toolik Lake" and (Fig. 6)

We have made this change.

L376: Section header using an unofficial "met" acronym for meteorological. I would replace for the full word or introduce an acronym and use more widely.

We have made this change.

L386: "…as high winds can and do remove…" should be "…as high winds can remove…"

We have made this change.

L402: "…and in a fully coupled lake-atmosphere system model the similar sensitivity experiments should provide different estimates." Should be, "…and in a fully coupled lake-atmosphere system, similar sensitivity experiments should provide different estimates"

We have made this change.

General:

Note that figure references sometimes switch styles from "Figure X" to "Fig. X", I guess this should be consistent.

We have checked these for consistency and made changes.

---

## Author Response (AR4)

GMD-2022-9
Thermal modeling of three lakes within the continuous permafrost zone in Alaska using LAKE 2.0 model
Response to Editor

28 Aug 2022
Author's Response:
Thank you for taking the time to handle our manuscript through the submission process over the last 8 months. We have updated the color schemes of Figures in Appendix D and Appendix E as requested by the editorial team. We have also lightly revised the abstract for readability and understanding. In particular we believe the uniqueness of our manuscript and modeling approach is highlighted in Lines 20-33 both in terms of differentiating the modeling from previous studies and in terms of the study results. Specifically the model parameter sensitivity analysis, meteorological variable perturbations, combination of local and remotely sensed input data, and validation using water depths at several depths over several years for three lakes differentiates our work from previous studies. If there are particular aspects of our study that you believe are not well represented in the abstract or if there are particular sentences that you believe need revision please let us know.

26 Aug 2022
Topical Editor decision: Publish subject to minor revisions (review by editor)
by Jinkyu Hong
Comments to the author:
I ask you to revise Abstract carefully because something important of your study is not clearly manifested. I feel that abstract can be improved further for better readibility.

Particularly, please revise the abstract by representing
1) why this lake modeling is different from previous arctic lake studies (modeling or experiment)
2) What are the unique findings of this study compared to other LAKE modeling studies.

Also, I am not a native speaker but I also feel some revision of sentences in Abstract for better understanding.

---

## Author Response (AR5)

GMD-2022-9
Thermal modeling of three lakes within the continuous permafrost zone in Alaska using LAKE 2.0 model
Response to Editor

9 Sept 2022
Author's Response:
We have added the missing DOI for the dataset reference. We believe both the dataset archive (ESS-DIVE, https://www.re3data.org/repository/r3d100000019) and code archive (github/zenodo) meet the journal requirements.

31 Aug 2022
Topical Editor decision: Publish subject to technical corrections
by Jinkyu Hong
Comments to the author:
I realized that code and data availability does not satisfy our journal requirement and something changed after my first request on this availability. Please read carefully our journal requirement and provide a permanent link such as zenodo for the observation data and code.